



# A proxy of subsurface Chlorophyll-a in shelf waters: use of density profiles and the below mixed layer depth (BMLD)

Arianna Zampollo[1,*], Thomas Cornulier[1], Rory O'Hara Murray[2], Jacqueline Fiona Tweddle[1], James Dunning[1], Beth E. Scott[1]

[1] School of Biological Sciences, University of Aberdeen, Aberdeen, AB24 2TZ, UK

[2] Marine Scotland Science, Aberdeen, AB11 9DB, UK

*Correspondence to*: Arianna Zampollo (zampolloarianna@gmail.com)

**Abstract**

Primary production dynamics are strongly associated with vertical density profiles, which dictate the depth of stratification and mixed layers. Climate change and artificial structures (e.g. windfarms) are likely to modify the strength of stratification and vertical distribution of nutrient fluxes, especially in shelf seas where fine scale processes are important drivers, affecting the vertical distribution of phytoplankton. To understand the effect of physical changes on primary production, identifying the linkage between density and phytoplankton profiles is essential. Here, the ecological relevance of eight density layers (DLs) obtained by multiple methods that define three different portions of the pycnocline (above, centre, below) was evaluated to identify a valuable proxy for subsurface Chlorophyll-a (Chl-a mg m$^{-3}$) concentrations. The associations of subsurface Chl-a with surface and deep mixing were investigated by hypothesizing the occurrence at the same depth of any DL and the maximum Chl-a layer (DMC) using Spearman correlation, linear regression, and a Major Axis analysis. Out of 1237 observations of the water column exhibiting a pycnocline, 78% reported DMCs above the bottom mixed layer depth (BMLD). This suggests that the BMLD is a boundary trapping Chl-a in shallow waters ($\leq$ 120 m). BMLD constantly described Chl-a vertical distribution despite surface mixing indicators, suggesting a significant contribution of deep mixing processes in supporting subsurface production under specific conditions (e.g. prolonged stratification, tidal cycle, and bathymetry). Using BMLD for defining subsurface Chl-a could be a valuable tool for understanding the spatiotemporal variability of Chl-a in shelf seas, representing a potential variable for ecological assessments.

**Keywords**

Climate change, BMLD, DMC, deep mixing, MLD, pycnocline, SCML, shelf sea



## 1. Introduction

As we begin to manage our oceans and coastal seas for more complex simultaneous uses, such as renewable energy developments, fishing and marine protected areas, it is becoming increasingly important understanding details of primary productivity at fine spatial scales. The temporal and sub-mesoscale (1 to 100 km) spatial patchiness of resources in coastal seas (Goebel et al., 2014; Martin, 2003) indicates a complex interplay of localized factors – such as circulation, river plumes, mixing and stratification – that seasonally characterize the different hydrodynamic regimes of the marine environment (Leeuwen et al., 2015; Cullen, 2015; Lévy et al., 2015). Besides very shallow waters, the vast majority of phytoplankton generally grows in stratified waters, where the pycnocline acts as a barrier against the mixing of the whole water column. The balance between stratification and mixing is determinant for phytoplankton flourishing in the euphotic zone, which, in shelf seas, fluctuates in time and space by the modulation of daily and biweekly tidal cycles (Klymak et al., 2008). Turbulent mixing of the water column requires energy sources from either the surface (e.g. wind stress, Ekman pump due to wind curl) or the deep waters (e.g. upwelling, eddy diffusion, tidal currents). Climate change is introducing variations in these physical factors, and therefore changes are expected in the overall mixing budget of our seas. Anomalies in circulation slow-down, sea-level rise, bottom and surface temperature have largely been described as driven by climate change in the last two decades (e.g. Bryden et al., 2005; Taboada and Anadón, 2012). However, their effects on the biological effects, especially those from the bottom-up regulation of primary production, are still partially understood (Lozier et al., 2011; Somavilla et al., 2017).

### 1.1 Subsurface chlorophyll-a maxima layers (SCMLs)

Many of the uncertainties of climate change impacts on primary production come from the difficulties in sampling the community composition and the total abundance throughout the whole water column. The vertical distribution of phytoplankton is one of the most relevant and challenging variables to sample in the marine environment. Contrary to the detection of surface blooms by satellite sensors, subsurface chlorophyll-a maxima layers (SCMLs) are often more difficult to describe and measure. SCMLs represent significant features in plankton systems (Cullen, 2015), they define where most of the bottom-up processes take place and can encompass more than 50% of the entire water column production (Weston et al., 2005; Takahashi and Hori, 1984). In the North Sea, the summertime (May-August) subsurface production contributes to the annual production of up to 20-50% and sustain the food chain in continental shelf waters during prolonged stratified conditions (Hickman et al., 2012; Richardson and Pedersen, 1998; Weston et al., 2005). Several studies linked the vertical distribution of maximum chlorophyll-a (Chl-a) to deep mixing processes (e.g. Brown et al., 2015; Richardson and Pedersen, 1998; Sharples et al., 2006; Zhao et al., 2019b) and identified the occurrence of deep Chl-a assemblages in the proximity of the pycnocline in shelf seas (e.g. Costa et al., 2020; Durán-Campos et al., 2019; Ross and Sharples, 2007; Sharples et al., 2001). Deep turbulent processes and stratification are notably linked in shelf seas, where the stratification is maintained by tidal cycles mixing the water column through horizontal circulation (Glorioso and Simpson, 1994; Loder et al., 1992; Sharples et al., 2006, 2001; Simpson et al., 1980; Zhao et al., 2019b). Maxima Chl-a have been identified at the base of the pycnocline in regions of strong tidal mixing at Georges Bank in August (Holligan et al., 1984) and within the western English Channel (Sharples et al., 2001). However, despite the clear linkage between SCMLs and stratified waters, the effects of climate change on ocean productivity has mainly been described in relation to the mixing processes above the pycnocline (within the upper mixed layer) (Somavilla et al., 2017), omitting the effects of deeper layer processes. In fact, studies of shelf waters suggest fast variations of the water column due to both surface and deep mixing processes, since the interplay of marine components occur within a thinner layer than in deep oceanic locations (Durski et al., 2004). The exclusive investigation of the surface mixed layer is likely to



bias the investigation of climate change impacts on primary production (abundance and distribution) in shallow sea/shelf regions and needs to be investigated further.

### 1.2 Mixed layer depth (MLD) and pycnocline characteristics

MLD has been largely considered as a central variable for understanding phytoplankton dynamics (Sverdrup, 1953), especially in oceanic sites, where several studies have investigated the ecological relevance of MLD on Chl-a vertical distribution (Behrenfeld, 2010; Carranza et al., 2018; Diehl, 2002; Diehl et al., 2002; Gradone et al., 2020), phytoplankton bloom events (Behrenfeld, 2010; Chiswell, 2011; D'Ortenzio et al., 2014; Prend et al., 2019; Ryan-Keogh and Thomalla, 2020, Sverdrup, 1953), and the effects of climate change (Somavilla et al., 2017). The nutricline exhibits positive

correlations with the upper mixed layer depth (Ducklow et al., 2007; Gradone et al., 2020; Holligan et al., 1984; Prézelin et al., 2000, 2004; Ryan-Keogh and Thomalla, 2020; Yentsch, 1974, 1980), and it has been generally associated with surface spring blooms or windstorm events (e.g. Banse, 1987; Carranza et al., 2018; Carvalho et al., 2017; Lande and Wood, 1987; Therriault et al., 1978). However, the effect of climate change on MLD and primary production is still an unsolved question (Lozier et al., 2011; Somavilla et al., 2017). The need for a much more detailed understanding of the

linkage between primary production, pycnocline characteristics and deeper turbulent processes is therefore a key area of research, especially in highly productive but spatially heterogeneous areas such as shelf waters and shallow seas.

The methods for identifying MLDs vary among marine environments, hydrodynamic regimes, or the spatial resolution of vertical profiles (Courtois et al., 2017; Lorbacher et al., 2006), because making use of a single method is difficult for spatiotemporally heterogeneous regions. MLDs are typically defined as the depth at which the density gradient exceeds

a specific value (threshold) (e.g. Kara et al., 2000), however this method presents issues in specific hydrodynamic conditions, such as over estimating MLD in regions with deep convection (e.g. subpolar oceans) (Courtois et al., 2017), or misidentifying water columns with a newly established shallow MLD over previous periods of stratification (Somavilla et al., 2017). Several sensitivity tests and comparisons have been conducted in oceanic waters (e.g. Carvalho et al., 2017; Courtois et al., 2017; González-Pola *et al.*, 2007; Holte and Talley, 2009), however, there are no standard methods of

investigation that adapts MLD's identification in shelf waters.

### 1.3 A new way forward: the base of the pycnocline (BMLD) as an ecological indicator of the vertical distribution of maxima Chl-a (DMC) in shelf waters

In this study, we proposed the adaptation of existing methods into a new algorithm able to cope with different vertical distributions of the density (therefore being able to deal with split pycnoclines and unusual shapes) to characterize the

heterogeneity of coastal/shelf/shallow waters and identify the depth between the pycnocline and i) the surface mixed layer depth (commonly known as "MLD", here renamed as *above mixed layer depth*, AMLD) and ii) the bottom mixed layer depth (BMLD). The method is validated for a region with 14 years of repeated surveys that covers a mosaic of habitats types in waters depths ranging from 20 to 120 m (north-western North Sea) driven by seasonal stratification, permanently mixed waters, regions of freshwater inputs and strong tidal mixing (Leeuwen et al., 2015). We investigated the ecological

relevance of both layers (AMLD and BMLD) in relation to the vertical distribution and abundance of Chl-a, and we compared the performance of these two proposed density layers to some of the other methods used in the literature. This new level of understanding is being developed in order to help the identification of key linkages between the physical environment and primary production at finer spatial scales ($\leq$ 1 km), which can be ecologically relevant for pressing issues in marine spatial management (e.g. seabed leasing for wind farms, locations of MPAs) and spatially explicit climate

change assessments.



## 2. Methods

Vertical samples of density and Chl-a (see Sect. 2.1) were used to characterize the relationship between subsurface Chl-a (described as abundance and vertical distribution, see Sect. 2.2) and stratification features (see Sect. 2.3 and 2.4) in shelf waters < 120 m. The most frequent methods used to identify vertical characteristics of density profiles (density layers –

DLs) (see Sect. 2.3) were compared to the proposed algorithm estimating the above and below limits of the pycnocline (AMLD and BMLD in Fig. 2). This algorithm is able to cope with density profiles having instability, or the pycnocline fractured in sections (see Sect. 2.4). Here, a new method identifying BMLD is proposed and its ecological application (together with other six DLs) is evaluated by comparing the vertical distribution of subsurface Chl-a during spring and summer (April-August) (see Sect. 2.5).

### 115 2.1 Physical and biological oceanographic samples

*In situ* summertime measurements of temperature, salinity, and fluorescence (a proxy of Chl-a abundance) were collected from a towed, undulating, CTD and a vertical CTD in the North Sea off the East coast of Scotland, UK, within the Firth of Forth (FoF) and Tay region for over 14 years (Fig. 1). A total of 426 profiles were gathered from 12 oceanographic campaigns carried out by Marine Scotland Science on board of the fisheries research vessels *Scotia* and *Alba na Mara*

(www.gov.scot/marine-and-fisheries). The data set comprises temperature, conductivity, and fluorescence measurements from the sea surface to the seabed (vertical resolution equals to 1 decibar) at a number of fixed stations sites from 2000 to 2014. Water samples were collected during each cast for calibration of the *in situ* sensor data. Temperature and conductivity measurements were quality controlled using the standard Marine Scotland Science editing procedure. The undulating CTD sampled the water column in June 2003 and July 2014 with a continuous vertical and horizontal

oscillation of the instrument throughout the water column from 2 to 5 m below the sea surface to 5 m from the seabed. Data were sampled at 1 second intervals, resulting in a vertical resolution comprising between 0.5 and 1 m, in water depths from 25 m to 115 m. More information about the oceanographic cruise in June 2003 are described in Scott et al. (2010), and the same method was used in July 2014. The processing of undulating CTD enabled to get 847 single profiles of the water columns. Overall, 1273 profiles from both types of sampling were extracted from April to August (April=3,

May=51, June=1115, July=66, August=38). *In situ* conductivity were converted first in Practical Salinity ($S_P$), then into Absolute Salinity ($S_A$), and *in situ* temperature was converted into Conservative temperature ($\Theta$) to calculate density ($\rho$) (*gsw_rho* function), using the TEOS-10 toolboxes (www.teos-10.org) within the *gsw* v1.0-5 package in R v3.6.3 (R Core Team, 2018).

### 135 2.1.1 Standardized vertical sampling for density and Chl-a

Since the proposed algorithm (described in Sect. 2.3) works with profiles at high vertical resolution (samples' distance is 1 m), the *in situ* casts were required to be standardized throughout the water column. Density ($\rho$) and Chl-a observations taken every 0.5 to 1 m were converted into measurements over regular depth intervals by smoothing and interpolating. This was achieved by fitting a generalized additive model (GAM) (Hastie and Tibshirani, 1990) using an adaptive spline

with $\rho$, or Chl-a, as a function of depth. The smoothing basis (knots) were selected in a range from 75% to 90% of the number of observations occurring within each profile. The obtained smooth function for each profile was used to predict $\rho$ and Chl-a at regular 1 m depth intervals. In order to maintain the same shape and values in each profile, the fitted curves at 1 m interval were visually checked by plotting the estimated and real profiles to visually identify possible errors. 15% of the shapes (*n*=89) were manually corrected by changing the number of knots in the GAM. The pre-processing analysis

resulted in advantageously eliminating multiple sampling at the same depth that would have affected the selection of





density layers' depths and maxima Chl-a, especially in transects with undulating CTD. The analyses were run in R v3.6.3 (R Core Team, 2018) using the *mgcv* v1.8-33 package.

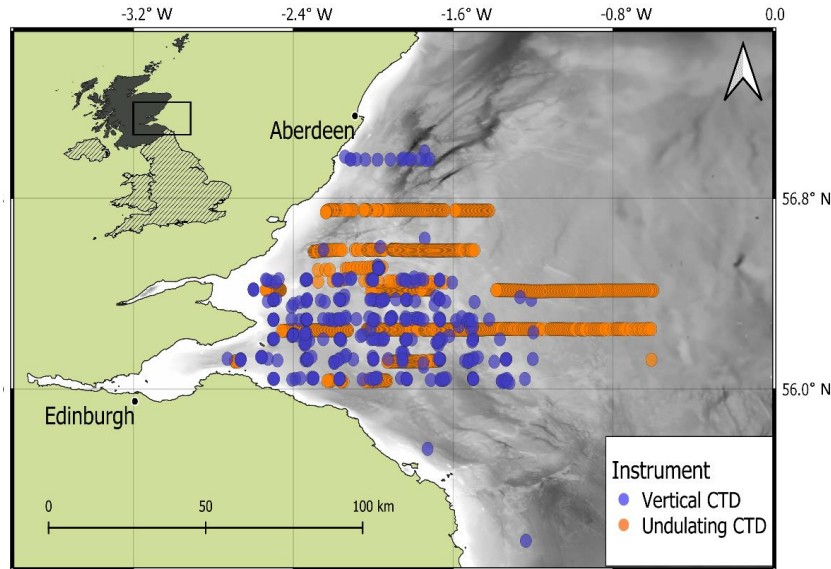

*Figure 1: Study area with the in situ surveys measured by an undulating CTD (orange dots) and a vertical CTD (blue dots). Land (green) and bathymetry (grey colour ramp) are pictured (ESRI 2020; EMODnet 2018)*


### 2.2 Subsurface Chlorophyll-a parameters

The depth of maximum Chl-a (DMC) was defined as the deepest maximum inflection point in the Chl-a profile standardized at 1 m sampling frequency (Carvalho et al., 2017; Zhao et al., 2019b), by using the adapted Chu and Fan (2011) method to measure the real angle instead of the tangent of φ (Eq. (1) and see details in Sect. 2.4). The automated

identification of DMC was checked manually with a visual inspection of each profile. The total amount of Chl-a were measured using trapezoidal integration (Walsby, 1997) throughout the water column (depth-integrated Chl-a) in R v3.6.3 (R Core Team, 2018).

The vertical distribution of Chl-a was classified into six most frequent vertical shapes according to the literature (Lavigne et al., 2015; Mignot et al., 2011; Uitz et al., 2006; Zhao et al., 2019a), using terminology adopted from Mignot et al., 2011

and Zhao et al., 2019. The profile was split in two sublayers, one above and one below the depth of maximum Chl-a (DMC), *upper* and *lower* sublayers (Fig. 2a grey solid line), and three equal sections were used to divide the difference between the minimum and maximum Chl-a values into three equal sections (Fig. 2a red dashed lines). The identification of the shapes was performed visually with the help of an automatic measuring of the ratio of observations in the three vertical sections within the *upper* and *lower* sublayers (Fig. 2). The few profiles with unclear subdivisions, or very

different shapes, were excluded from the dataset (which only represented 2% of the data).

First, the gaussian shapes, which were not determined by the ratio of observations within each section, have been pulled from the dataset and gathered into two shapes, the "Narrow-SCM" and "Wide-SCM", since the profiles exhibited two main widths of standard deviations of Chl-a from DMC. The Narrow-SCM shape is defined by the decrease of Chl-a from DMC within a limited range of depths (3-10 m) (Fig. 2a), while Wide-SCM shape is characterized by the equal decrease





of Chl-a within a wide range of depths above and below DMC, whose gaussian curvature often covers the whole water
column (Fig. 2b). The "SCM-HCU" shape exhibits a high ratio of Chl-a in the first *lower* and second *upper* sections (Fig.
2c), while the "SCM-HCL" shape is characterized by a high ratio in the first *upper* and second *lower* sections (Fig. 2d).
The "HCL" and "HCU" shapes are defined by the section with the highest ratio of Chl-a in the *lower* sublayer: HCL is
characterized by most of the observations within the third section (Fig. 2e), while the HCU exhibits a high number of
observations within the first section (Fig. 2f).

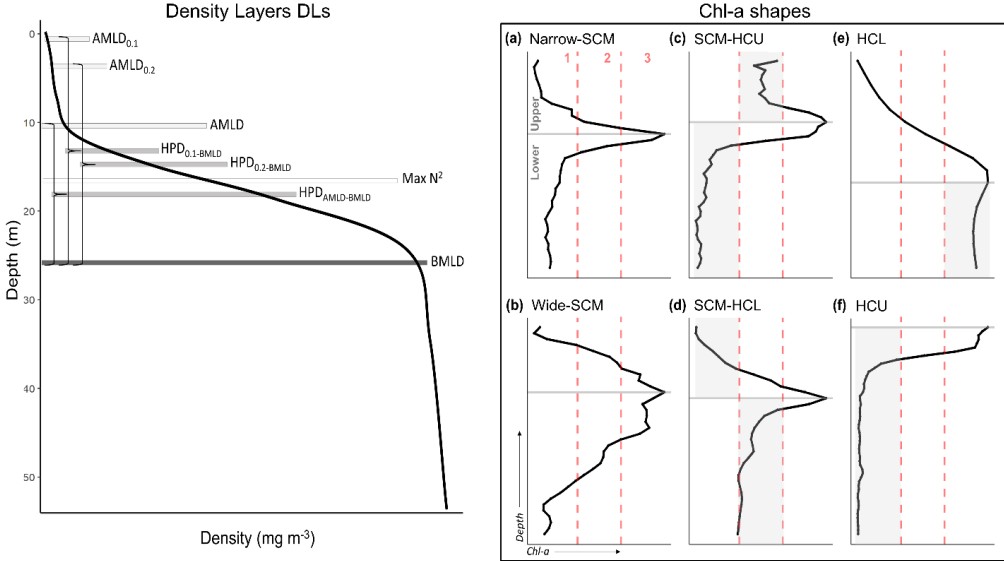

*Figure 2: The eight density layers (DLs) are reported for a generic density profile (on the left), together with an
example for each of the six (plots a-f) identified Chl-a shapes (on the right). On the density profile, the curly brackets
define the halfway depth (HPD) between AMLD's indicators (AMLD$_{0.1}$, AMLD$_{0.2}$, AMLD) and BMLD. The Chl-a
shapes are split into the upper and lower sublayers at the DMC (horizontal solid grey line) (a). The vertical lines
indicate the limits of sections 1, 2 and 3 (dashed red lines) (a) that were used to identify the type of shape. The grey
shaded squares represent the sections with the highest ratio of Chl-a determining SCM-HCU and SCM-HCL, HCL and
HCU.*

*Table 1: Table with the abbreviations used in the paper.*

| Abbreviation | Description |
|---|---|
| SCML | *Subsurface Chlorophyll-a maximum Layer* |
| Chl-a | *Chlorophyll-a (mg m$^{-3}$)* |
| DMC | *Depth of maximum Chlorophyll-a (m)* |
| DL | General abbreviation for a *density layer* (e.g. AMLD, BMLD, HPD, or Max N$^2$) *(m)* |
| MLD | General expression for *Mixed layer depth (m)* |
| AMLD | *Above mixed layer depth,* or starting point of the pycnocline *(m)* |
| BMLD | *Below mixed layer depth*, or ending point of the pycnocline *(m)* |
| HPD | *Halfway pycnocline depth,* or centre of the pycnocline *(m)* |



| | |
|---|---|
| *Max N²* | Maximum water buoyancy frequency (N²) *(m)* |


### 2.3 Common methods identifying Density Layers (DLs)

among the methods used to detect density layers in coastal and oceanic waters, three approaches were selected to define mixing and buoyancy features in the sampled profiles.

The AMLDs are typically defined as MLD in the literature and represent the depth at which the density gradient exceeds

a specific value (threshold method) (e.g. Kara et al., 2000). The threshold is typically selected among a range of values previously tested in the literature (from 0.0025 to 0.125 kg m$^{-3}$) (summarized in Holte and Talley, 2009; Lorbacher et al., 2006; Montégut et al., 2004) and measured as the difference ($\Delta\rho_z = |\rho_z - \rho_{ref}|$) between a certain sampling depth ($z$) and a reference density value ($\rho_{ref}$), which can be the density at the surface, 10 m depth, or a consecutive point (e.g. $z+1$). In this study, two density gradients (0.01 and 0.02 kg m$^{-3}$) have been measured as the difference between two consecutive

points in the profile ($\Delta\rho_z = |\rho_z - \rho_{z+1}|$) and named as $AMLD_{0.01}$ and $AMLD_{0.02}$.

Since previous studies identified DMCs in the proximity of the centre of the pycnocline (HPD), we investigated the relationship between DMCs and three different HPDs measured as the halfway depth between the base of the pycnocline (BMLD, see Sect. 2.4) and $AMLD_{0.01}$, $AMLD_{0.02}$ and adjusted AMLD (the last described in Sect. 2.4), and named $HPD_{0.01-BMLD}$, $HPD_{0.02-BMLD}$, and $HPD_{AMLD-BMLD}$ (Fig. 2).

Moreover, the association of maximum buoyancy frequency squared (Max N²) with DMC and Chl-a abundance has been investigated since several studies reported positive correlation at oceanic (e.g. Martin et al., 2010; Schofield et al., 2015; Carvalho et al., 2017; Courtois et al., 2017; Baetge et al., 2020) and shelf waters (Lips et al., 2010; Zhang et al., 2016). For each profile, the depth of Max N² has been selected from N² profiles (Fig. 2) computed by *gsw_Nsquared* function (*gsw* v1.0-5 package) in R v3.6.3 (R Core Team, 2018), which is based on absolute salinity and conservative temperature

with respect to pressure following the most recent version of the Gibbs equation of state for seawater in TEOS-10 systems (Intergovernmental Oceanographic Commission, 2010). The magnitude of N² quantifies the stability of the water column and pinpoints the stratified layers where the energy required to exchange water parcels in the vertical direction is maximum (Boehrer and Schultze, 2009).

### 2.4 AMLD and BMLD detection

Theoretically, the layers between the pycnocline and a mixed vertical region above and below the pycnocline are depths showing a large change in the density gradient. The surface mixed layer depth (AMLD) and the mixed layer depth below the pycnocline (BMLD) are both transient layers from a mixed to a stratified vertical region occurring at the beginning and end of the pycnocline. The threshold methods (see Sect. 2.3) delineate an AMLD's identification based on the principle that the mixed layer at the surface is characterized by a variance of Δρ close to zero. They assume that the

pycnocline is the portion of the water column with a large density gradient Δρ that separates two portions of mixed waters (above and below it) exhibiting a low and similar Δρ. These assumptions may not always hold, and we found that identification failure can occur when the upper mixed layer is heterogeneous, with nested sub-structures such as small re-stratification at the surface followed by a small mixed layer before the pycnocline (Fig. A1e in Appendix A), or when the pycnocline is fractured in chunks (Fig. A1f in Appendix A). These conditions are difficult to isolate using the maximum

angle (Chu and Fan, 2011) and threshold methods. In this paper, the AMLD's definition does not assume that the surface mixed layer is fully mixed with a Δρ close to zero for the whole portion of the water column, and it identifies AMLD regardless any *a priori* threshold. It also picks up the shallowest and deepest limits of the pycnocline by excluding middle breaks of the pycnocline, allowing the identification of unconventional density vertical distribution. Instead, here, the





definition of AMLD and BMLD are based on common conventions: small and similar variations in the density gradient
within the mixed layer, above and below the pycnocline; the pycnocline is enclosed by mixed portions of the water column
above and/or below it exhibiting a significant variation of the density gradient; the depth with the largest density is
pinpointed independently from a fixed gradient (Chu and Fan, 2019, 2011; Holte and Talley, 2009).

AMLD and BMLD have been identified developing an algorithm based on Chu and Fan (2011) framework to produce a
method able to cope with various density profiles exhibiting a pycnocline (examples in Fig. A1 in Appendix A). The
algorithm's sequence identifies the depth with the largest density gradient between a mixed and a stratified layer using i)
an adaptation of the maximum angle method (Chu and Fan, 2011) and ii) a cluster analysis on the density gradient ($\Delta\rho_z = |\rho_z - \rho_{z+1}|$) (diagram of the algorithm in Fig. 3a). The method is designed to work with equal, high-resolution, intervals
of density values (z) in the profiles. In order to distinguish AMLD from BMLD, their selection is achieved by splitting
the observations throughout the profile into two distinct groups, *Split1* and *Split2* (Fig. 3b and Fig. 3c), each one
respectively used to identify AMLD and BMLD. *Split1* includes the density values within the first observation close to
the surface ($z_l$) and two measurement intervals δ (here 1 m) above BMLD ($z_{BMLD} – 2δ$), while *Split2* extends from 2δ
above the depth halfway through the ρ range ($0.5\Delta\rho = ((\rho_{max} - \rho_{min})/2) - 2$) up to the depth at which the total number of
points from the surface to the bottom amounts up to 90% of the entire profile ($z_{0.9\Delta\rho} = 90\%$ of $\frac{n}{1}z$). Since *Split1* is based
on BMLD, the algorithm identifies AMLD after BMLD.

For all depths between $z_1$ and $z_{0.9\Delta\rho}$, the angle φ has been measured at $z(x, y)$ (where x is the density and y is depth)
between two vectors (V1, V2) fitting a linear regression ($y \sim x$) each. The two vectors have been calculated using 2δ
before and after each observation (z) (V1 = from [$z – 2$] to z, and V2 = from z to [$z + 2$]) (Fig. 3b and Fig. 3c). Although
Chu and Fan (2011) suggested to measure the tangent of the angle between V1 and V2 (φ), we encountered some issues
identifying BMLD in those profiles that decreased in density below the BMLD (Fig. A1d, Appendix A). Therefore, the
algorithm has been improved by calculating the angle φ. Since the slope (or angular coefficient, $\beta$) of a linear regression
is the tangent of the angle between the line and the x-axis, the angle φ was obtained from two angles extracted from the
coefficients measured by V1 and V2 according to the sign of $\beta$: i) positive $\beta$ (see example in Fig. 3d, angle $\tau$ and the
orange vector) refers to the angle between the vector and the horizontal plane with $y$ equal to the intercept (α), or ii)
negative $\beta$ (see example in Fig. 3d, angle $\omega$ and the blue vector) refers to the angle between the vector and the vertical
plane with $x = 0$. The angle φ at each observation ($\varphi_z$) is measured by summing up, or subtracting, the angles derived
from the coefficients, $\beta_1$ and $\beta_2$ for V1 and V2, according to their partial contribution to φ, which can be summarized
under four different conditions:

$$\varphi_z = \begin{cases} \text{atan}(|\beta_1|) + \text{atan}(|\beta_2|), & \beta_1 > 0 \text{ and } \beta_2 > 0 \\ \text{atan}(|\beta_2|) - \left(\frac{\pi}{2} - \text{atan}(|\beta_1|)\right), & \beta_1 > 0 \text{ and } \beta_2 < 0 \\ \text{atan}(|\beta_1|) + \left(\frac{\pi}{2} - \text{atan}(|\beta_2|)\right), & \beta_1 < 0 \text{ and } \beta_2 > 0 \\ |\text{atan}(|\beta_1|) - \text{atan}(|\beta_2|)|, & \beta_1 < 0 \text{ and } \beta_2 < 0 \end{cases} \quad (1)$$

where atan() refers to the arctangent of the coefficients $\beta_1$ and $\beta_2$.

Up to this stage, the algorithm selects AMLD and BMLD on the adapted maximum angle method (Chu and Fan, 2011).
However, the exclusive use of the maximum angle method would have biased the selection due to local variation and
instability conditions of the water column (Fig. A1b, c, e, f in Appendix A). Therefore, a K-Mean cluster analysis (Lloyd,
1982) was adopted in the algorithm to improve the selection of the pycnocline limits by adding a further step of selection
on the 3 and 5 largest φ for AMLD and BMLD, respectively. Since the transition from surface mixing layer to the



pycnocline is sharper than that one from the pycnocline to the bottom mixing layer, the number of φ candidates is higher in BMLD than in AMLD selections. The cluster analysis classifies the density gradient at depth ($\Delta\rho_z = |\rho_z - \rho_{z+1}|$) into groups (see below), assuming that $\Delta\rho_z$ values within a mixed layer would belong to a unique cluster.

AMLD's selection is made amongst the 3 largest φ, and the first $\varphi_z$ amongst the descendent ordered candidates meeting the following conditions was assigned as AMLD: i) the observations ($z$) within the mixed water column belong to the
same cluster classification (CC), the candidate $\varphi_z$ must have $CC_z = CC_{z+1}$ and $CC_z \neq CC_{z1}$ (CC at surface $z_1$), ii) and $\Delta\rho_{z-1} < \Delta\rho_z$. In AMLD's selection, the $\Delta\rho_z$ is grouped in two clusters since we would expect two main variations of $\Delta\rho$ in *Split1*: a small gradient on the surface mixed section and a bigger one at the pycnocline due to stratification. The same approach has been adopted for BMLD's identification amongst the 5 largest φ, although the inclusion of three clusters instead of two improved the performance of the algorithm since the region of the water column transiting from the
pycnocline to the bottom mixed layer is smoother than in AMLDs (e.g. Fig. A1b in Appendix A). The first $\varphi_z$ amongst the descendent ordered candidates meeting the following conditions was selected as BMLD: i) $CC_z = CC_{z-1}$ and $CC_z \neq CC_{z0.9\Delta\rho}$ (CC at the z=0.9Δρ), and ii) $\Delta\rho_z < \Delta\rho_{z-1}$. Adding the conditions controlling for a similar classification of $\Delta\rho_z$ at depths above AMLD and below BMLD resulted in decisive outcomes, correctly identifying the mixed layers within those density profiles having a pycnocline fractured in chunks with different or similar gradients. However, when the conditions
associated with clustering were not found among the candidates φ, the algorithm was not necessary and therefore the simplest methods were adopted to select i) AMLD with a threshold gradient $\Delta\rho_z > 0.02$ mg m$^{-3}$, and ii) BMLD as the largest φ (Fig. 3a). The algorithm was developed in R v3.6.3 (R Core Team, 2018), and the K-mean density was calculated using the *kmeans* function using Lloyd (1982) algorithm (*stats* package).



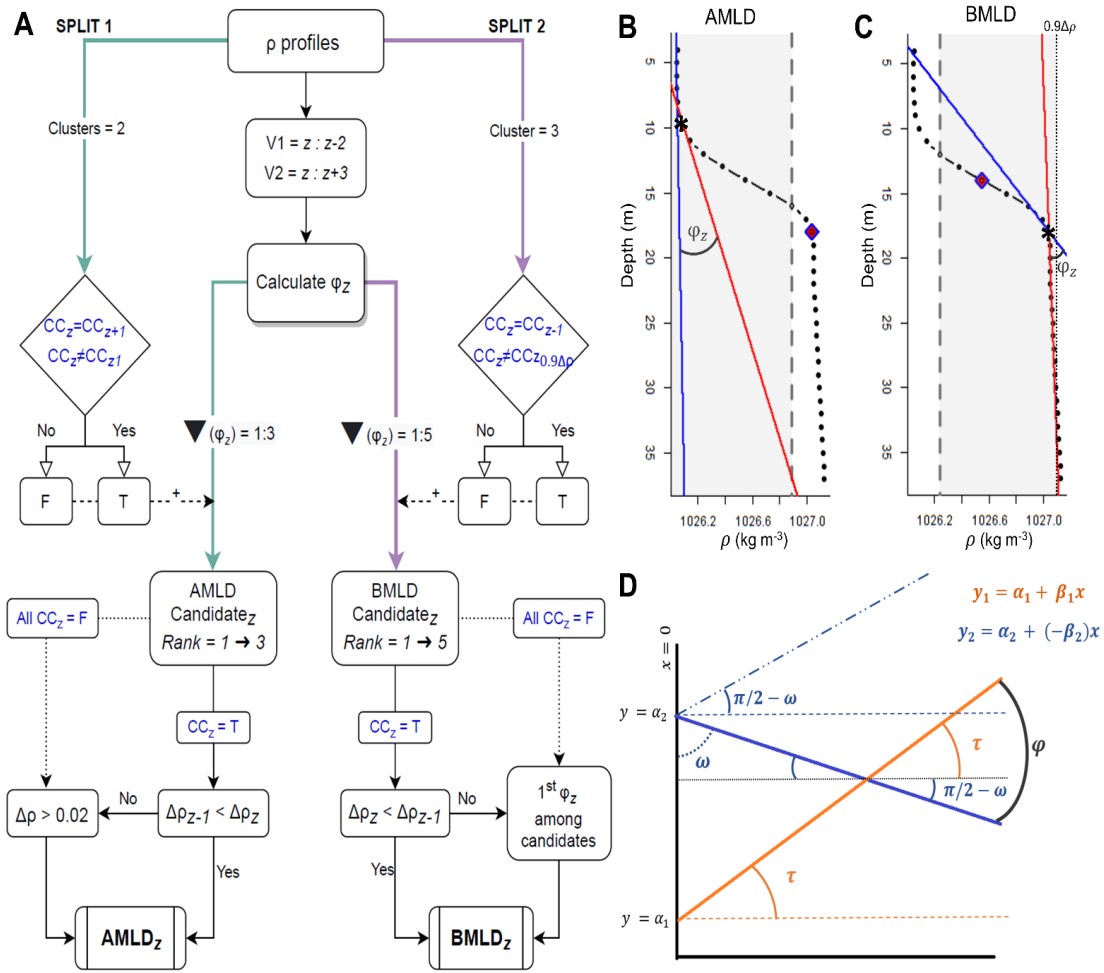

*Figure 3: main steps of AMLD and BMLD selection: (a) diagram of the algorithms, where green arrows belongs to Split1 and purple arrows to Split2, text in blue is the portion of the algorithm relying on cluster analysis (K-mean), "F" and "T" are the results, false and true, of the conditions expressed in the rhombuses. The φ is measured for each observation (z), and the largest (3 for AMLD and 5 for BMLD) φ are considered as candidates of AMLD and BMLD. The candidates are descendent ordered (Rank 1 → 3 or Rank 1 → 5) and the first candidate meeting the other*

*conditions will be identified as AMLD or BMLD. If any candidate meets the conditions, the original methods are used (threshold method > 0.02 and maximum angle φ). (b) and (c) are plots of the same density profile representing the attributes used in the algorithm: grey region includes the observations (black dots) used to identify AMLD and BMLD, which extends in (b) from the surface to two depths above BMLD (purple rhombus), and in (c) from two depths above the middle of the pycnocline (purple rhombus) to 0.9Δρ. AMLD and BMLD are reported by a black star in (b) and (c)*

*respectively. In (b) and (c), the vectors V1 (blue line) and V2 (red line) are drawn for each z (black star) and φz is reported. Plot (d) shows of one of the four conditions reported in Eq. (1) measuring φ: V1 (orange line) with a positive slope (β1) and V2 (blue line) with a negative slope (β2).*





### 2.5 Evaluating the association between density layers and subsurface Chl-a

The ecological relevance of each density layer (DL) was evaluated by comparing their coincidence with the depth of
maximum Chl-a (DMC) (e.g. DMC = BMLD) and the predictability of DMC ($y$) from each DL ($x$). The coincidence and
the prediction of DMCs from a density characteristic are important tools for understanding the processes driving
subsurface concentrations and identifying a valuable proxy for modelling analyses or controlling uncertainty in net
primary production estimates.

In this study, we evaluated the coincidence of the DMC with eight investigated density layers ($AMLD_{0.01}$, $AMLD_{0.02}$,
AMLD, BMLD, $HPD_{0.01-BMLD}$, $HPD_{0.02-BMLD}$, $HPD_{AMLD-BMLD}$, and Max $N^2$, Fig. 2, described in Sect. 2.3 and 2.4) using
Spearman's rank correlation coefficient ($\rho_S$) and a Major Axis (MA) line fitting, and the prediction of DMC from DL by
performing a linear regression model (LM). The Spearman's coefficient (Eq. (2) in Table 2) assesses a monotonic linear
relationship with values ranging between -1 and +1, which refer to a perfect negative or positive correlation between two
variables. Besides the strength of the linear relationship defined by $\rho_S$, we focused on evaluating the linear relationship
between DMC and each DL using 3 different linear models $y = \alpha + \beta x$: 1) alpha and beta estimated by linear regression
(Eq. (4) in Table 2); 2) alpha and beta estimated by major axis line fitting; and 3) the one-to-one line with alpha and beta
fixed at 0 and 1 respectively (Eq. (4) in Table 2). The MA is largely used to investigate how one variable scales against
another by accounting for errors from both directions ($x$ and $y$) and measuring the residuals perpendicular to the line
(details in the review Warton et al., 2006). Therefore, the aim of MA is not to predict the $y$-variable, however evaluating
the proximity of the coefficients of the estimated MA line ($\alpha$ and $\beta$) to the scenario in which DL equals DMC. The
coincidence of each DL and DMC was summarized by reporting the $\alpha$ and $\beta$ MA coefficients, which are here
hypothesized to reflect the one-to-one line (intercept $\sim 0$, slope $\sim 1$) if the DMC is aligned with the DL in question.

Since the identification of a proxy for subsurface Chl-a represents a useful tool for correctly assessing the abundance and
the variations of primary production, we investigated the power of prediction of DMC from each DL by measuring the r-
squared ($R^2$) from i) an ordinary least square to estimate parameters from the observations in a linear regression (Eq. (3)
in Table 2), and ii) the one-to-one linear regression (which has been forced with the intercept through the origin and a
slope equal to 1, Eq. (4) in Table 2). The formulae used to calculate the coefficient of determination $R^2$ for the one-to-one
($R_0^2$) and empirical ($R_{em}^2$) LMs have been summarized in Eq. (3) and  Eq. (4) (Table 2).

*Table 2: Formulae for estimating the bivariate line-fitting. Spearman's rank correlation coefficient ($\rho_S$), coefficient of*
*determination $R^2$ for testing the one-to-one linear regression ($R_0^2$) (e.g. DMC $\sim$ BMLD) and the empirical linear*
*regression ($R_{em}^2$).*

|  | **Formula** | | **Purpose** |
|---|---|---|---|
| $\rho_S$ | $\dfrac{\sigma_{xy}}{\sigma_x \sigma_y}$ | (2) | Estimate the strength of the relationship between $x$ and $y$ |
| $R_{em}^2$ | $1 - \dfrac{SS_{RES}}{SS_{TOT}} = 1 - \dfrac{\sum_{i=1}^{n}(y_i - \hat{y}_i)^2}{\sum_{i=1}^{n}(y_i - \bar{y})^2}$ | (3) | Measure the variation in $y$ that is explained by $x$ in a LM |
| $R_0^2$ | $1 - \dfrac{SS_{RES}}{SS_{TOT}} = 1 - \dfrac{\sum_{i=1}^{n}(y_i - x_i)^2}{\sum_{i=1}^{n}(y_i)^2}$ | (4) | Measure the variation in $y$ that is explained by $x$ in a one-to-one LM |




*Notation: $\sigma_{xy}$ is the covariance of x and y, $\sigma_x$ and $\sigma_y$ are standard deviations, n is the number of observations of $x$ and*

*y, $y_i$ is $DMC_i$, $\bar{y}$ is the average of DMCs, and $x_i$ is the density layers related to DMC in each regression (e.g. DMC*

*~ BMLD). $SS_{RES}$ is the residual sum of squares, $SS_{TOT}$ is the total sum of squares.*

In the LM, $R^2_{em}$ was calculated using the typical formula with the residual sum of squares ($SS_{RES}$) as the square of the difference of $y$ and $\hat{y}$ (estimated $y$ from the model) (Eq. (3) in Table 2). In the one-to-one LM, the $SS_{RES}$ in $R^2_0$ was adapted by replacing $\hat{y}$ with $x$ (Eq. (4) in Table 2), since the values of $x$ and $y$ are assumed to be equal in the one-to-one line regression and the difference between them should be zero. The two $R^2$ differ also for the denominator $SS_{TOT}$, which

is the sum of squares about the average of the explanatory variable in $R^2_{em}$ and the sum of squares of the DMC values since in $R^2_0$ the value of DMC and DL equals.

Since the $SS_{TOT}$ adopted in the two formulae is different, the proportion of explained DMCs' variance by each DL can be compared only within each linear regression rather than across the one-to-one and empirical regressions. Therefore, the power of prediction among DLs was discussed in within each type of LM.

### 3. Results

The presented algorithm identifying for AMLD and BMLD was applied to the 1273 profiles exhibiting a pycnocline (see Sect. 3.1), whose associations with DMCs (and with the other density layers – $AMLD_{0.01}$, $AMLD_{0.02}$, $HPD_{0.01\text{-}BMLD}$, $HPD_{0.02\text{-}BMLD}$, $HPD_{AMLD\text{-}BMLD}$, and Max $N^2$) are described for the whole dataset (see Sect. 3.2) and for each Chl-a vertical distribution (see Sect. 3.3).

### 3.1 Identification of AMLD and BMLD

The above mixed layer depth (AMLD) and the below mixed layer depth (BMLD) were identified by merging existing methods into an algorithm able to process density profiles with a 1 m sampling resolution. The algorithm was applied to the 1273 profiles exhibiting a pycnocline with heterogeneous vertical distributions, e.g. having a small re-stratification at the surface followed by a mixed layer before the pycnocline, or a pycnocline fractured in sections (examples of density

profiles in Fig. A1, Appendix A).

Here, the identifications of AMLD and BMLD did not assume that the mixed layer has a density gradient ($\Delta\rho$) close to zero (e.g. threshold methods). Instead, the occurrence of a layer (the pycnocline) having $\Delta\rho$ at any observation $z$ ($\Delta\rho_z = |\rho_z - \rho_{z+1}|$) pointedly different from that within the above and below (mixed) layers, is assumed. Therefore, the algorithm pinpoints the transition from the mixed layers to the pycnocline based on similar variations in $\Delta\rho$ within the

mixed layer and within the pycnocline. As Fig. 2 shows, the algorithm was created to identify i) AMLD as the depth between a surface mixed layer having $\Delta\rho$ similar among observations and a layer (pycnocline) exhibiting an increasing $\Delta\rho_z$ after AMLD, and ii) BMLD as the depth at which $\Delta\rho_z$ is smaller than at the pycnocline and consistently similar among observations up to the seabed. This identification does not consider the pycnocline as a layer with a constant $\Delta\rho$ throughout its whole extension, since the pycnocline can include a small mixed layer (Fig. A1a, e, f in Appendix A) or

presents different density gradients (stratified layers) within it (Fig. A1b and c in Appendix A). Therefore, the AMLD represents the last depths up to which the $\Delta\rho$ is consistently small from the surface to the pycnocline, while the BMLD is the first depth after a layer with large $\Delta\rho$ from which the density gradient is consistently small down to the seabed (or the deepest observation). In the algorithm, the similarity amongst $\Delta\rho_z$ was measured using a cluster analysis (see Sect. 2.4), which defines the main conditions controlling the selection of AMLD and BMLD by hypothesising that the mixed layer

(up to AMLD or from BMLD) must have density gradients belonging to the same cluster. However, in specific conditions





the algorithm failed to correctly identify AMLD and BMLD and classified the two limits of the pycnocline within it (Fig. A1, Appendix A). The selection was considered to have failed when the AMLD and BMLD were selected ≥ 2 m (2 observations) above or below the mixed layer depth. Major errors in identifying AMLD (6.76% of the profiles) and BMLD (4.32%) occurred in density profiles with a high number of observations within the transition from the mixed layer to the pycnocline, where $\varphi_z$ was similar amongst several observations and the cluster analysis was identifying the gradients close to the end of the pycnocline as belonging to the mixed layer (e.g. Fig. A1 a-c, Appendix A). The number of candidates appeared to be sensitive to the sampling frequency and the thickness of the transition regions (AMLD-pycnocline, pycnocline-BMLD). Therefore, it is important to highlight the sensitivity of this method to the rate of change of the gradients at AMLD and BMLD (a large rate of change is preferred), and the sampling frequency at the transition between the pycnocline and the above and below mixed layers. The algorithm did not correctly identify AMLD in profiles without a surface mixed layer, and a shallow pycnoclines that comprised two different gradients (Fig. A1c). In this case, the cluster analysis split $\Delta\rho$ into two groups, although they belong to the same pycnocline. Other errors were related to profiles having a pycnocline split into two by a small mixed layer within a depth range > 4 m (4 observations) (Fig. A1e). Overall, the identification of BMLD performed better than AMLD's, although it could not deal with profiles having less than 4 observations throughout the pycnocline (in this study thickness of the pycnocline < 3 m). This condition occurred due to the location of the *Split2* (which is necessary to distinguish BMLD's from AMLD's selection) i) at depths above AMLD (misidentifying AMLD as BMLD) or ii) too close to BMLD (missing enough observations to fit properly V1). The algorithm always correctly selected BMLD in profiles that have the lowest densities below the BMLD (Fig. A1d).

### 3.2  DMC association with different characteristic of the density profile

The depth of maximum Chl-a (DMC) was compared to the location of eight features of the density profiles (DLs described in Sect. 2.3 and 2.4, Fig. 2) that are summarised in surface mixed layer depth ($AMLD_{0.01}$, $AMLD_{0.02}$, AMLD), bottom mixed layer depth  (BMLD), the centre of the pycnocline ($HPD_{0.01-BMLD}$, $HPD_{0.02-BMLD}$, $HPD_{AMLD-BMLD}$) and the depth of maximum buoyancy frequency squared (Max $N^2$) to evaluate i) the strength of a positive linear relationship between each DL and DMC, and ii) the power of prediction of DMC by each DL.

All the methods classifying the surface mixed layer ($AMLD_{0.01}$, $AMLD_{0.02}$ and AMLD) showed the location of these density layers to generally be shallower than DMCs (Fig. 4 a-c, Table 3) with a rare coincidence of their vertical distribution (from 0.39% to 1.73% of the profiles, Table 3). In particular, the two thresholds used to identify AMLD (0.01 and 0.02) exhibited the lowest Spearman correlation amongst all DLs, with $AMLD_{0.01}$ having almost a zero correlation to DMCs ($\rho_S$ = -0.01) and a null explanation of the DMC's variability in the empirical linear regression ($R^2_{em}$ = 0.00). The Major Axis analysis identified intercept and slope values in $AMLD_{0.01}$ and $AMLD_{0.02}$ almost perpendicular to the *y*-variable due to the strong presence of DMCs in deep waters. Although a clear subsurface aggregation of max Chl-a occurs below the surface mixed layer (Fig. 4c, the AMLD measured by the algorithm (Sect. 2.4) showed a better correlation with DMC than $AMLD_{0.01}$ and $AMLD_{0.02}$, with a positive linear relationship between the two variables and a greater explained variance of DMC by the one-to-one and empirical linear regressions (Table 3).

Max $N^2$ is the density layer performing least well after AMLDs in predicting DMCs, although it showed the highest percentage of coincidence with DMCs (13.51% of the profiles, Table 2). Similar to AMLDs, DMCs have been recorded in 64.96% of the profiles at layers deeper than Max $N^2$, indicating that max Chl-a area located in waters below surface mixing, at stratified regions within the pycnocline. Overall, the centre of the pycnocline (HPDs) distributed close to DMCs, with $HPD_{AMLD-BMLD}$ exhibiting the highest performance: the highest correlation to DMCs ($\rho_S$ = 0.56), and the highest explained DMC's variance from the one-to-one ($R^2_0$ = 0.90) and empirical ($R^2_{em}$ = 0.31) LMs (Table 3). The





location of DMCs is highly related to HPD$_{AMLD-BMLD}$, although only 4.63% of the profiles presented DMCs and HPD$_{AMLD-BMLD}$ at the same depth (Table 3). Many profiles exhibited DMC deeper than HPD$_{AMLD-BMLD}$ (78.69%), of which 81.53% distributed DMCs above BMLD (hence, between HPD$_{AMLD-BMLD}$ and BMLD).

The below mixed layer depth, BMLD, exhibited a reverse condition compared to the other density layers by encompassing 78.32% of DMCs in waters above it (Table 2). BMLDs is the second variable after HPD$_{AMLD-BMLD}$ with the highest correlation to DMCs ($\rho_S = 0.55$), it is distributed at the same depth of DMCs in 7.86% of the profiles and linearly predicted the location of maxima Chl-a in both one-to-one and empirical linear regressions (Table 2). BMLD exhibited MA coefficients ($\alpha = 0.60$ and $\beta = 0.82$) close to the hypothesized one-to-one fitting-line ($\alpha = 0$ and $\beta = 1$), indicating a good approximation of DMCs at BMLD.

The overall distribution of DMCs is discernible mainly (> 95.84% of profiles) below the surface mixed layers (AMLDs' indicators), within the deepest half of the pycnocline (between HPD$_{AMLD-BMLD}$ and BMLD) and it is bounded for 78.32% of the observations above the BMLD. However, although DMCs generally reflect the region with the highest concentration of Chl-a throughout the water column, the vertical concentration of phytoplankton can vary in the proximity of DMCs and accumulate mainly above or below it (Fig. 4). The ecological relevance of the density layers has therefore
been investigated in comparison with different Chl-a profile shapes (Fig. 5).



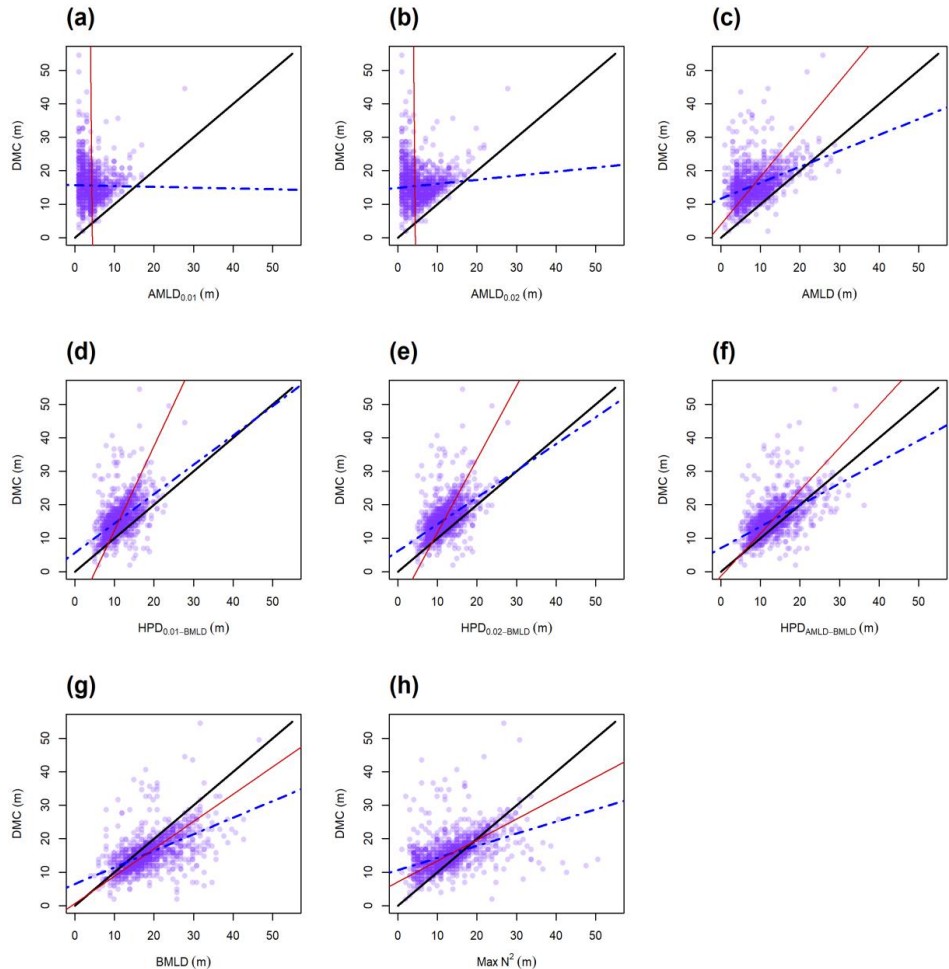

*Figure 4: Scatterplots of DMC and the eight DLs (a-h). The lines refer to the one-to-one linear regression (LM) (solid black), the Major Axis analysis (MA) (solid red), the empirical LM measured from the observations (DMC ~ DL) (dot-dashed blue).*


*Table 3: Statistical parameters and profiles' percentages having DMCs above (>), at the same depth (=), or below (<) each DL.*

| DL | $\rho_S$ | $\alpha$ | $\beta$ | $R_0^2$ | $R_{em}^2$ | DMC > DL | DMC = DL | DMC < DL |
|---|---|---|---|---|---|---|---|---|
| $AMLD_{0.01}$ | - 0.01 | 543.35 | -124.26 | 0.40 | 0.00 | 99.53 | 0.39 | 0.08 |
| $AMLD_{0.02}$ | 0.08 | -43.72 | 11.35 | 0.47 | 0.01 | 99.45 | 0.31 | 0.24 |
| AMLD | 0.41 | 4.01 | 1.42 | 0.69 | 0.17 | 95.84 | 1.73 | 2.44 |
| $HPD_{0.01-BMLD}$ | 0.52 | -12.81 | 2.52 | 0.86 | 0.27 | 90.18 | 1.81 | 8.01 |
| $HPD_{0.02-BMLD}$ | 0.52 | -10.20 | 2.19 | 0.87 | 0.27 | 86.41 | 3.77 | 9.82 |
| $HPD_{AMLD-BMLD}$ | 0.56 | 1.31 | 1.28 | 0.90 | 0.31 | 74.86 | 4.63 | 20.50 |
| BMLD | 0.55 | 0.60 | 0.82 | 0.87 | 0.31 | 13.83 | 7.86 | 78.32 |





| Max N$^2$ | 0.45 | 7.06 | 0.63 | 0.84 | 0.20 | 64.96 | 13.51 | 21.52 |

### 3.3 Chl-a vertical distribution in relation to density layers

425 Since hydrodynamic and biological conditions shape Chl-a differently throughout the water column through processes such as resuspension, passive drift, and mortality (i.e. zooplankton grazing in stratified and stable waters), Chl-a can have very different vertically distributions in relation to DMC values (Fig. 5).

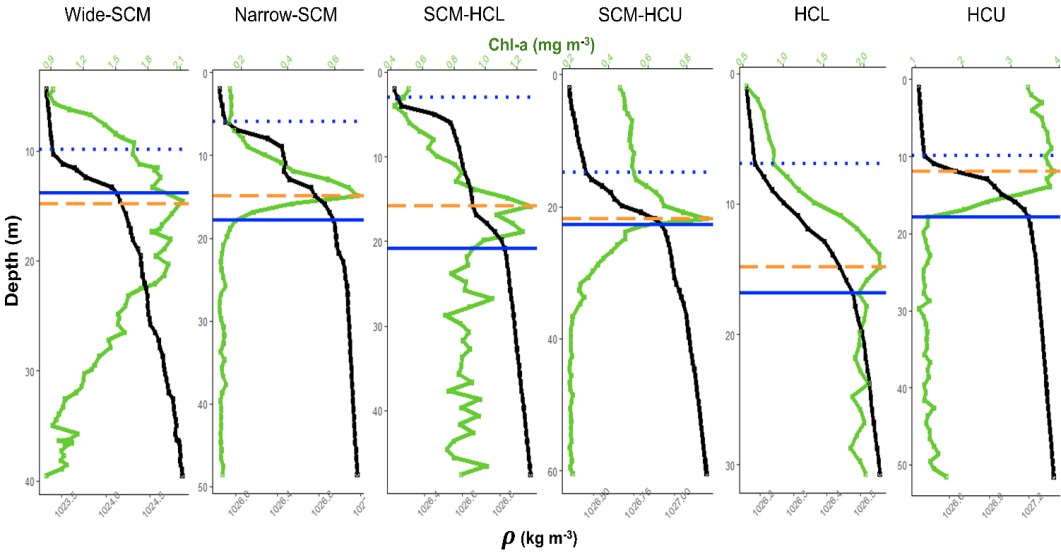

*Figure 5: Example vertical distribution of Chl-a (green solid line) and density (black solid line). The horizontal lines*
430 *indicate BMLD (blue solid), AMLD (blue dotted), and DMC (yellow dashed).*

The depth-integrated Chl-a was standardized ("standardized depth-integrated Chl-a") by the number of 1 m observations above and below four DLs (AMLD, HPD$_{AMLD-BMLD}$, BMLD and Max N$^2$) and values were compared (Table 4). AMLD and HPD$_{AMLD-BMLD}$ were selected amongst the density layers indicating the surface mixed layer and the centre of pycnoclines due to their better correlation to DMC (see Sect. 3.2). The amount of Chl-a (mg) at each meter depth above
435 and below the four density layers is reported in Fig. A2 (Appendix A).

Following the results in Sect. 3.2, a large portion of Chl-a was measured at depths below AMLD, HPD$_{AMLD-BMLD}$ and Max N$^2$ (Table 4), where DMCs also occurred. HPD$_{AMLD-BMLD}$ and Max N$^2$ delimit almost three times the amount of Chl-a at depths included from these vertical locations to the seabed as compared to the concentrations at the surface. A reverse condition is exhibited by Chl-a distributing above and below BMLDs: the standardized depth-integrated Chl-a is higher
440 above than below BMLDs, although the amount of phytoplankton in the deepest layers is still comparable (the difference between surface-BMLD and BMLD-seabed is 42.80 mg m$^{-1}$) (Table 4) (Fig. A2 in Appendix A shows the full distribution of Chl-a values at the 1 m sampling resolution).

It is therefore sensible to infer the distribution of DMCs, and the largest portion of phytoplankton at depths enclosed within the stratified region (AMLD – BMLD), to be mainly in the second half of the pycnocline (HPD$_{AMLD-BMLD}$ –




BMLD). At the same time, a reasonable amount of Chl-a distributes below the pycnocline (BMLD), especially in SCM-HCL and HCL shapes (Fig. 5 and 6).

Table 4: Values of depth-integrated Chl-a (mg) standardized by its range of vertical distribution (m) (Total Chl-a biomass (mg)/depths (m)) above and below the four density layers. These values are also reported in Fig. A2 (Appendix A).

| DL | Standardized depth-integrated Chl-a above DL (mg m$^{-1}$) | Standardized depth-integrated Chl-a below DL (mg m$^{-1}$) |
|---|---|---|
| AMLD | 172.97 | 971.12 |
| HPD$_{AMLD-BMLD}$ | 366.07 | 859.27 |
| BMLD | 615.92 | 658.72 |
| Max N$^2$ | 372.90 | 848.14 |


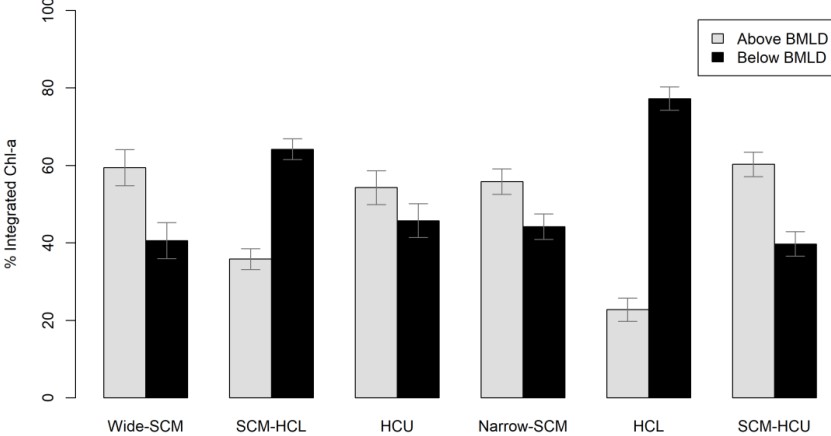

*Figure 6: Bar plot of the median percentage of Chl-a above (light grey) and below (black) BMLD for each Chl-a shape. Grey bars refer to standard error.*

### 3.3.1    DLs associated with Chl-a shapes, with a focus on BMLD

Since BMLD exhibited the clearest pattern in defining the vertical distribution of Chl-a, further investigations have been focused on understanding the relationship between BMLD and Chl-a. The percentage of depth-integrated Chl-a above and below BMLD was measured for each profile and the median values are reported in Fig. 6. HCL and SCM-HCL shapes exhibited a high concentration of Chl-a at depths below BMLD, while SCM-HCU, Narrow-SCM, Wide-SCM and HCU are characterized by large concentrations between the sea surface and BMLD.

A distinct pattern of deep Chl-a is visible in HCL shapes, where 77.24% of the total Chl-a was recorded below BMLDs (Fig. 6), and 87.14% of the profiles (*n*=70) reported DMCs in deep mixed waters (Table 5). HCL shape were significantly recorded at shallow bathymetry (≤ 63.15 m) (Wilcoxon test on bathymetry values at HCL profiles and all the other



profiles, W = 70534, $p < 0.00$) and exhibited an exceptionally high concentration of Chl-a at DMCs amongst all the other profiles (Wilcoxon test, W = 57303, $p < 0.00$) (Fig. A4b in Appendix A). HCL shapes exhibited a high correlation to

BMLD than to the other density layers (Table 5, Tables A1-A7 in Appendix A), although the coincidence of DMC with BMLD occurred only in 1.43% of the profiles. BMLD exhibited a better performance amongst the other density layers in predicting DMCs from both one-to-one and empirical linear regressions. The MA analysis reported slope values < 1 in all the shapes except HCL, which has the highest $\beta$ coefficient and the most negative intercept (Fig. 7 and Table 5).

The SCM-HCL exhibits, with HCL shape, the greatest linear relationship between DMC and BMLD, showing the highest

coincidence of BMLDs and DMCs (10.86% of 405 profiles, Table 5). Amongst all the investigated density layers, DMCs in SCM-HCL locate at depths very close to the base of the pycnocline (Fig. 7 and Table 5) although a large portion of the depth-integrated Chl-a (64.17%) occurred between BMLD and the seabed (Fig. 6). BMLD shows the best performing empirical and one-to-one linear regressions amongst all the Chl-a shapes (Table 5).

The absence of a solid pattern in Wide-SCM shape reflects its extensive range of depth at which Chl-a distributes

throughout the water column. In Wide-SCM shapes, HPDs' indicators exhibited the highest correlation to DMCs amongst all the density layers (Tables A4-A6 in Appendix A), especially $HPD_{0.1-BMLD}$ and $HPD_{0.2-BMLD}$ (Fig. 7) (MA coefficients $\alpha$ and $\beta$ close to 0 and 1 respectively), while the percentage of profiles with DMC equal to BMLD appeared higher (7.20%) than HPDs. The one-to-one and empirical linear regressions similarly report weaker predictability of DMCs from BMLD than the other Chl-a shapes.

Since the Narrow-SCM shape typically describes the aggregation of Chl-a within a thin layer of the water column (3-10 m), DMCs are identified between AMLD and BMLD in 83.91% of the profiles ($n$=404), with 55.82% of the total Chl-a between the sea surface and BMLD (Fig. 6). The MA analyses indicate BMLD and $HPD_{AMLD-BMLD}$ as the closest DLs to DMC amongst all the shapes (Fig. 7), whose $\alpha$ and $\beta$ values measured almost 0 and 1 respectively ($\alpha = -0.26$ and $\beta = 0.87$ for BMLD, $\alpha = 0.22$ and $\beta = 1.13$ for $HPD_{AMLD-BMLD}$). All the DLs except for $AMLD_{0.01}$ and $AMLD_{0.02}$ efficiently

predicted DMCs from both one-to-one and empirical linear regressions (Table A1-A7 in Appendix A).

The SCM-HCU shape exhibits the highest percentage of depth-integrated Chl-a from the sea surface to BMLD (60.27%, Fig. 6), with 91.02% of the profiles ($n$ =245) have the DMC above the base of the pycnocline. The shape showed the highest coincidence of DMCs at Max $N^2$ (16.88% of the profiles) amongst all the density layers (Tables A7 in Appendix A), although the MA coefficients exhibit a low co-occurrence of DMC at Max $N^2$ (Fig. 7). The MA analyses indicate

BMLD and $HPD_{AMLD-BMLD}$ as the closest DLs to DMC (Fig. 7); however, the empirical and one-to-one linear regressions with BMLD and the surface mixing layers performed less well than HPDs' indicators and Max $N^2$.

For the HCU shapes, the Spearman coefficient shows a low positive correlation between DMCs and DLs, except for the upper mixed layer indicators (AMLD, $AMLD_{0.01}$ and $AMLD_{0.02}$) that occurred at the same depth of DMCs for almost 17% of the profiles ($n$=24, Table A1-A3 in Appendix A). Similarly, DMCs occur at the same depth of Max $N^2$ for 16.67%

of the profiles with a relatively high Spearman coefficient ($\rho_S = 0.55$), although Max $N^2$ exhibits the lowest $R_0^2$ (-0.11) and a low $\beta$ from the Major Axis analysis ($\beta = 0.34$, Table A7 in Appendix A). The same condition refers to BMLD, which predicts only -0.04 of DMC's variance ($R_0^2$) in HCU shapes and reports DMCs to be always shallower than BMLD (100%, Table 5). Amongst the DLs, BMLD is the density layer with the closest MA coefficients to the ideal co-occurrence of DMCs at BMLD (Fig. 7).




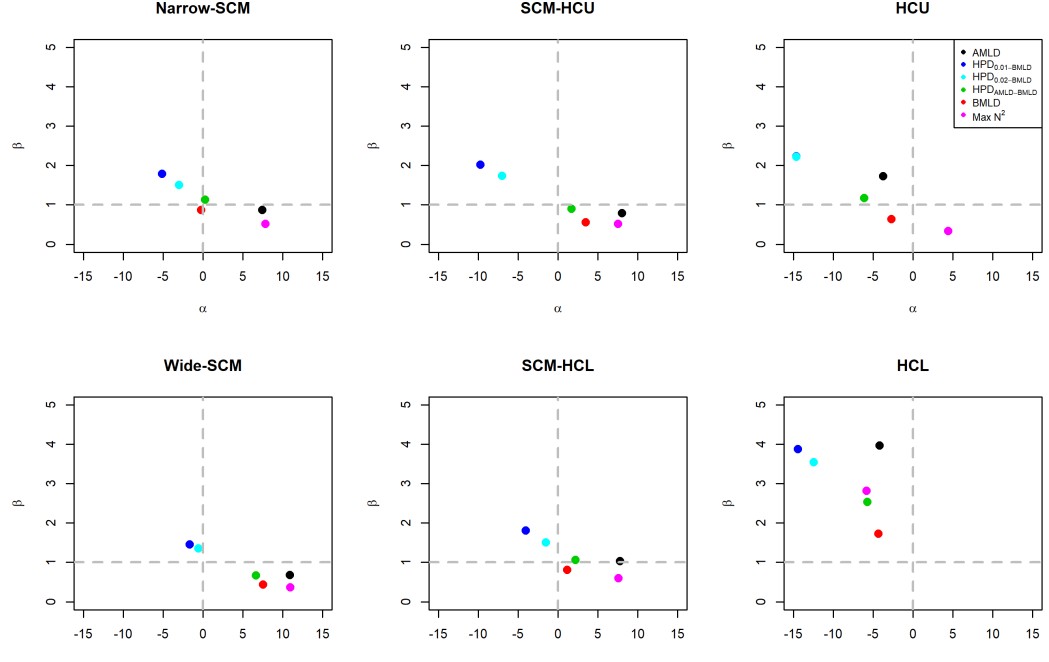

*Figure 7: one plot for each Chl-a shape reporting the MA coefficients (α and β, values reported in Table 5 and Tables A1-A7 in the Appendix A) for six DLs (AMLD$_{0.1}$ and AMLD$_{0.2}$ were excluded due to their large values visible in Fig. 4 a-c). In each plot, the dashed grey lines (α=0 and β=1) crosses where the DL is hypothesized to occur at the same depth of DMC. Top-right and bottom-left panels (as defined by the dashed grey lines) represent systematic over- and under-*

*estimation respectively, while top-left is under-estimation of the lower values, and bottom-right is over-estimation of the lower values.*

*Table 5: Statistical parameters and profiles' percentages having DMCs above (>), at the same depth (=), or below (<) the BMLD.*

| DL = BMLD | | | | | | | | | |
|---|---|---|---|---|---|---|---|---|---|
| Chl-a shape | $n$ | $\rho_S$ | $\alpha$ | $\beta$ | $R_0^2$ | $R_{em}^2$ | DMC > DL | DMC = DL | DMC < DL |
| Wide-SCM | 125 | 0.58 | 7.51 | 0.44 | 0.79 | 0.33 | 16.00 | 7.20 | 76.80 |
| SCM-HCL | 405 | 0.78 | 1.16 | 0.81 | 0.94 | 0.61 | 13.33 | 10.86 | 75.80 |
| HCU | 24 | 0.55 | -2.73 | 0.65 | -0.04 | 0.30 | 0.00 | 0.00 | 100 |
| Narrow-SCM | 404 | 0.77 | -0.26 | 0.87 | 0.95 | 0.59 | 8.42 | 7.67 | 83.91 |
| HCL | 70 | 0.70 | -4.36 | 1.73 | 0.83 | 0.49 | 87.14 | 1.43 | 11.43 |
| SCM-HCU | 245 | 0.50 | 3.48 | 0.56 | 0.77 | 0.25 | 2.86 | 6.12 | 91.02 |

**4. Discussion**





In stratified waters the vertical distribution of Chl-a is partially defined by physical factors, whose contribution to stabilize the stratification and mixing rate throughout the water column varies across hydrodynamic regions over time (Leeuwen et al., 2015). Stratification and mixing characterize the heterogeneous physical environment in shallow and shelf waters. The example here of the North Sea demonstrates the interplay of static (e.g. topography, shelf edge, position of river outflow) and dynamic variables (e.g. wind stress, tidal phases, amount of river outflow, convection or eddy activities), which go on to influence the whole food web at the local scale. The combination of static, dynamic and biological factors (e.g. grazing, Benoit-Bird *et al.*, 2013) induces phytoplankton communities to adopt different vertical distributions that can be ecologically important at small scales (Scott et al., 2010; Sharples et al., 2013, < 1 km). Understanding the relationship between Chl-a and vertical density at a fine spatial scale is essential to assess the effects of variations in physical processes due to large scale factors (e.g. stratification strength or changes in mixing rate due to wind and tidal renewable energy extraction). In order to identify the vulnerable link of primary production with variations of the hydrodynamic regimes, key physical proxies consistently associated with the different conditions of subsurface Chl-a (shapes) need to be investigated. The differences in the association of DLs, Chl-a shapes and depth-integrated Chl-a with DLs are discussed in the context of previous studies in order to understand the underlying conditions and propose a valuable tool to help predict subsurface Chl-a at finer scales.

### 4.1 Ecological relevance of AMLD in defining DMCs: valuable in HCU shape

Oceanic sites exhibit phytoplankton blooms within the upper mixed layer (e.g. Behrenfeld, 2010; Costa et al., 2020; Somavilla et al., 2017) to coincide with AMLDs' vertical fluctuations due to e.g. windstorm events deepening the pycnocline into nutrient-enriched waters (Detoni et al., 2015; Carranza et al., 2018; Höfer et al., 2019; Montes-Hugo et al., 2009). In this study, all the investigated surface mixed layers' indicators ($AMLD_{0.01}$, $AMLD_{0.02}$ and AMLD) weakly predicted DMC, reporting low linear correlations for all Chl-a shapes (Tables A1-A3 in Appendix A). The algorithm used in this study has reported an overall high performance in predicting the location of DMCs in HCU shape, which exhibited the shallowest DMCs (on average $9.74 \pm 6.66$ m standard deviation). HCU shapes represent ephemeral surface blooms in shelf waters, whose DMCs resulted mainly at layers ≤ the upper mixed layer depths. According to literature (Carranza et al., 2018; Zhao et al., 2019a), HCU showed the highest correlation to the upper mixed layer depth by exhibiting the largest percentage of DMCs above: $AMLD_{0.1}$ and $AMLD_{0.2}$ in 4.17% of the profiles, and AMLD in 25%. The AMLD identified by the proposed algorithm tested as the best variable in predicting most of DMCs in the one-to-one ($R_0^2 = 0.76$) and empirical ($R_{em}^2 = 0.34$) linear regressions, while BMLD accurately always defined the deepest boundary of DMCs in the observations (Table 4).

Since AMLD has been largely considered as a central variable for understanding phytoplankton dynamics (Sverdrup, 1953), it has been investigated in relation to climate change to infer possible significant changes in the amount, spatial distribution and phenology of oceanic primary production (Boyd et al., 2015; Montes-Hugo et al., 2009; Somavilla et al., 2017; Prend et al., 2019; Richardson and Bendtsen, 2019; Schmidt et al., 2020). However, the effect of climate change on AMLD and primary production is still an unsolved question (Lozier et al., 2011; Somavilla et al., 2017). The unclear effects of climate change on AMLD and primary production might be related to i) the difficulties in measuring the amount of subsurface Chl-a and its little association to satellites' observations at the sea surface (Baldry et al., 2020; Erickson et al., 2016; Lee et al., 2015), and ii) the exclusive investigation of the effects of surface mixing processes on primary production (e.g. temperature, wind-induced mixing) by neglecting deep processes that are responsible for the pycnocline's stability (Dave and Lozier, 2015, 2013; Lozier et al., 2011; Somavilla et al., 2017). As described above, the AMLD is informative for surface concentrations (HCU shapes), but it may not be biologically relevant for subsurface Chl-a that are




maintained at the pycnocline by deep turbulent mixing. The need for a much more detailed understanding of the linkage between subsurface Chl-a, pycnocline characteristics and deep turbulent processes is therefore a key subject, especially in highly productive but spatially heterogeneous areas such as shelf waters and shallow seas.

## 4.2 Association of subsurface Chl-a with DLs

The observations in the FoF and Tay region with a wide variety of characteristics of shallow seas, confirmed the subsurface presence of maxima Chl-a between April and August, with DMCs distributing on average (± standard deviation) at depths (m) equal to 17.22 ± 4.95 in Wide-SCM, 15.08 ± 4.47 in SCM-HCL, 14.82 ± 3.29 in Narrow-SCM, 22.69 ± 10.91 in HCL, and 15.17 ± 4.16 in SCM-HCU. A recent study in the German Bight described DMCs located mainly at the centre of the pycnocline and the overall amount of Chl-a at depths distinctly lower than the surface mixed

layers (Zhao et al., 2019a). The vertical distribution of DMCs at BMLDs appeared to be correlated to the bathymetry by exhibiting DMCs closer to BMLDs at bathymetry comprised from, approximately, 40 to 70 m (in Narrow-SCM, SCM-HCL and Wide-SCM shapes), DMCs deeper than BMLD mainly in shallow waters (in HCL shapes, generally < 60 m), and DMCs above deep BMLD towards deeper waters (in SCM-HCU and HCU shapes, generally from 30 to 100 m) (Fig. A5 A in Appendix A). Previous studies identified a similar pattern in shallow waters where DMCs were mainly recorded

at or below the base of the pycnocline (here BMLD) (Barth et al., 1998; Durán-Campos et al., 2019; Holligan et al., 1984; Zhao et al., 2019a). The link between bathymetry and Chl-a shapes, and the association of DMC with BMLD become important in those regions where bathymetry plays an important role in defining the location of commercial interests such as in the FoF and Tay region, location of several offshore wind farms (www.marine.gov.scot). The installation feasibility will allow the deployment of wind turbines in water depths ranging from 41 to 58 m above the lowest astronomical tide

(LAT) (www.marine.gov.scot), where reliable environmental impact assessment, able to estimate the indirect effects in a holistic way, are required.

### 4.2.1   Stable Chl-a shapes

Narrow- and Wide-SCM shapes can be considered as relatively stable vertical distribution of Chl-a since they occur during stable stratified conditions (Cullen, 2015; Carranza et al., 2018). DMCs at the pycnocline (between AMLD and

BMLD) have been consistently recorded in Narrow-SCM profiles within a pycnocline's width about 8.81 ± 3.83 m on average (± standard deviation) (Fig. A4c in Appendix A). The location of DMC at the pycnocline in Narrow-SCM is regulated over time by upward nutrient-enriched fluxes entering the pycnocline from deep waters (Pingree et al., 1982; Rosenberg et al., 1990). In the Skagerrak strait between Denmark and Norway, deep SCMLs were recorded at a nutricline (rate of change in nitrate and phosphate) located below the base of a shallow pycnocline (< 15 m) (Bjørnsen et al., 1993).

A low number of Narrow-SCM profiles exhibited DMCs deeper than BMLDs (8.42%), while this condition (DMC > BMLD) was more evident in Wide-SCM profiles (16%) having a thicker and variable pycnocline (on average 12.76 ± 6.85 m) than Narrow-SCM profiles. The higher variability in the location of DMCs and BMLDs in Wide-SCM ($R_0^2 = 0.79$) than Narrow-SCM ($R_0^2 = 0.95$), and the extended distribution of Chl-a throughout the whole water column in Wide-SCM might reflect a limited erosion of Chl-a by mixing and grazing above and below the pycnocline. Overall, the deep

distribution of DMCs, and most of the depth-integrated Chl-a, in the proximity of the centre and the base of the pycnocline suggests the maintenance of subsurface Chl-a within shelf waters through the regulation of nutrient supply by deep physical processes.

### 4.2.2   Transient Chl-a shapes





Besides the stable Narrow- and Wide-SCM shapes, the other profiles (HCL, HCU, SCM-HCU and -HCL) have been described in the literature as transient frames either from a stratified to a mixed water column or vice versa. Carranza et al. (2018) described two vertical distributions of Chl-a (from HCU to SCM-HCU) occurring from a mixed to stratified phase of the water column, indicating the ephemeral persistence of these shapes in the marine environment, eventually developing the typical (Narrow- or Wide-) SCM shapes. Although SCM-HCU and HCU profiles develop DMCs above AMLDs in Carranza et al. (2018), the observations in the FoF and Tay region reported DMCs deeper than AMLDs'

indicators in > 62.50% of HCU profiles and > 91.43% of SCM-HCU (Fig. 6). Similarly to SCM-HCU, SCM-HCL might reflect the transition from stratified to mixed conditions, where phytoplankton cells concentrated at SCMLs are re-suspended and diluted in deep layers due to an increasing tidal current (Zhao et al., 2019a). Beside SCM-HCU and -HCL might reflect different transitions between mixing and stratified conditions, only BMLD appeared a consistent proxy in defining the limit above which the DMCs have developed and, hence, is further discussed.

*SCM-HCU shape*

In SCM-HCU profiles, the DMCs occurred at Max $N^2$ at a larger percentage (15.10% of the profiles) than the other density indicators. The depth of Max $N^2$ is a less turbulent region where the energy to exchange parcels in the vertical is maximum (Boehrer and Schultze, 2009), and it is frequently used to identify the upper mixed layer (e.g. Carvalho et al., 2017). The location of DMCs at Max $N^2$ in SCM-HCU profiles might reflect the distribution of phytoplankton within a less turbulent region where nutrient particles, which have been resuspended by mixing, can persist for longer time periods. The mild

turbulent layer at Max $N^2$ would therefore represent a hot spot of nutrients reached by resuspended phytoplankton cells, while strong mixing processes still undergoing above and/or below it, or diluted gradients of phytoplankton and nutrients throughout the water column, would avoid the creation of highly productive subsurface patches. Although the depth of Max $N^2$ resulted in SCM-HCU being more informative than BMLD, DMCs exhibited a clear pattern by distributing

shallower than BMLDs in 91.02% of the profiles and representing the deepest limit up to which DMCs distributed. Overall, Max $N^2$ exhibited higher percentages of coincidence with DMCs (13.51% of 1273 profiles) than other DLs (Table 3), although the linear correlation ($\rho_S$), the MA coefficients and the one-to-one linear regression $R_0^2$ described a low association of DMCs with Max $N^2$ compared to HPDs' indicators and BMLD (Table 5 and Tables A4-A7 in Appendix A). However, the use of Max $N^2$ in summertime shelf waters to infer the depth of subsurface Chl-a patches in a one-to-

one fitting-line (DMC = Max $N^2$) may lead to underestimate the amount of Chl-a in the whole water column, as the amount of standardized depth-integrated Chl-a below Max $N^2$ is almost three times higher than above it (Table 4 and Fig. A2 in Appendix A).

*SCM-HCL shape*

SCM-HCL exhibited a greater association of DMCs with BMLDs than HPDs' indicators or Max $N^2$, with the largest

coincidence of DMC at BMLD (10.86% of the profiles) and α and β coefficients from the Major Axis analysis close to a one-to-one fitting-line (Table 5, Fig. 7). It was not the aim of this study to assess if the transient phase is taking place either from mixed to stratified waters or vice versa, although the closer proximity of DMCs to BMLDs than Max $N^2$, and the higher percentage of DMCs below Max $N^2$ (73.09% of the profiles against 13.33% DMCs below BMLD) might indicate the erosion of a stable pycnocline where DMCs previously developed (transition from a stratified to a partially

mixed water column). In the German Bight, 76% of SCM-HCL profiles presented high Chl-a at the base of the SCMLs, suggesting a possible erosion of the subsurface layer from the bottom due to strong tidal currents (Zhao et al., 2019a). The physical factors developing SCM-HCL might not cause the mixing of the whole water column and, instead, sustain an indispensable upward flux of nutrients into the enduring pycnocline, where e.g. dinoflagellates are able to compete




successfully in slightly turbulent conditions (< 0.1 mm s⁻¹) (Ross and Sharples, 2007). Therefore, the erosion as well as the resuspension of previously sinking phytoplankton cells and nutrients can maintain the proximity of DMCs at BMLDs. Although SCM-HCL appears to be a transient shape with a short-life (Zhao et al., 2019a), it has been widely encountered (*n*=405) during summer in the FoF and Tay region, and therefore its permanency might occur at a temporal scale (e.g. spring-neap cycle) that allows phytoplankton to counteract the dispersion of the gradients. Moreover, the large amount of diluted Chl-a in deep waters (64.17% of depth-integrated Chl-a below BMLD, Fig. 6) might be crucial in maintaining primary production at the subsurface over the summer, since deep mixing processes eroding and sustaining Chl-a at BMLD would contribute also to reducing the overlap between SCMLs and predators (Behrenfeld, 2010).

### 4.2.3 HCL shape and BMLD in shallow waters

The opposite condition is found in HCL profiles, where DMCs have been identified in deep layers below BMLD in 87.14% profiles (Table 4). The large portion of deep Chl-a, which is typical in HCL shapes, is described in the literature as primary production trapped in deep waters by a surface layer with a low diffusivity (e.g. pycnocline) (Jones et al., 1998; Zhao et al., 2019a). Besides the potential physical drivers inducing Chl-a below BMLD (77.24% of depth-integrated Chl-a is below BMLD, Fig. 6), deep Chl-a is probably accumulated due to the slowdown of the current at the seabed (Neill and Hashemi, 2018). In particular significantly more HCL profiles (results in Sect. 3.3.1) have been recorded in shallow waters (from 22.45 to 63.15 m, on average 30.77 ± 11.59 m, Fig. A5b in Appendix A) as well as in other studies (Jones et al., 1998; Huisman et al., 2002; Zhao et al., 2019a), where a compatible amount of light and suspended sediments can sustain phytoplankton growth throughout most of the water column (Huisman et al., 2002). Although sinking rates have been described as the main driver of Chl-a distribution below BMLD (Jones et al., 1998; Huisman et al., 2002; Zhao et al., 2019a), the density at DMCs showed a similar range (1021 – 1028 kg m⁻³) to the other Chl-a shapes exhibiting deep DMCs below BMLD (Wide-SCM, SCM-HCL) (Fig. A4a and Table A8 in Appendix A report no significant differences between these shapes), suggesting that hydrodynamic drivers (e.g. deep turbulent nutrient-enriched fluxes) might have more of an effect on Chl-a profiles than density on sinking rates. Another characteristic of the HCL shape is the exceptionally high concentration of Chl-a at DMCs than all the other profiles (results in Sect. 3.3.1, and Fig. A4b in Appendix A). It is evident that HCL profiles occurred at stratified conditions, probably when the tidal speed was slow enough to allow the stratification to persist (Zhao et al., 2019a) below a thin pycnocline (on average 8.82 ± 5.19 m, Fig. A4c in Appendix A) able to trap down a significantly large amount of Chl-a over shallow regions. Therefore, the provenance of high Chl-a at depth in shallow regions (≤ 63 m) might be due to the passive drift and accumulation by horizontal tidal currents in shallow waters, or the sinking combined with resuspension and active photosynthesis. Overall, high concentrations of Chl-a below the pycnocline represented a distinct pattern in shallow waters, revealing the sensitivity of these regions to further changes in the stratification strength or mixing at a small scale (< 1 km) of the water column due to manmade structures (e.g. renewable deployments).

### 4.3 The role of BMLD in further climate change investigations

Regions with large and deep phytoplankton concentrations are highly important for absorbing and sinking atmospheric carbon dioxide and represent a biological pump of carbon sequestration (Boyd et al., 2015). The correct estimation of the abundance of subsurface primary production is therefore highly important in investigating climate change implications in the marine environment. The exclusion of subsurface Chl-a in shelf waters is estimated to undervalue the total productivity of up to 10%-40% (Sharples et al., 2001). This amount of underestimation and lack of understanding of exact mechanisms for changes in vertical location of density and Chl-a would strongly affect the wider scale assessment of climate change impacts as well as the finer scale of manmade structures on the biological functionality of a certain region. The location





of the BMLD was overall the best variable constantly informing about the locations of DMCs throughout the water
670   column. However, we want to highlight that a minimum of 39% of depth-integrated Chl-a is found within waters below
the BMLD and this represents a high proportion of potential primary production that needs to be considered. In terms of
abundance of primary production, the Northeast Atlantic shelves exhibited a summertime reduction of Chl-a in the last
60 years leading to significant impacts on the food web in the North Sea (Capuzzo et al., 2018; Schmidt et al., 2020). In
particular, the intensified stratification caused an effective reduction in nutrient supply at the surface with the
consequential starvation and change of phytoplankton communities (e.g. Bindoff et al., 2019; Boyd et al., 2015; Schmidt
et al., 2020). The isolation of surface waters from deep nutrient-rich waters may explain the distribution of phytoplankton
at the subsurface, especially in the proximity of BMLD, which represents the limits up to which the deep nutrient-enriched
fluxes distribute and allows phytoplankton to grow in a region with low turbulence (Bopp et al., 2013; Boyd et al., 2015).
Not only the stratification strengthening but also the vertical distribution of BMLD and the upward fluxes, up to the
pycnocline may either redistribute food patches at major depths, together with the deepening of BMLD, and causing an
overall reduction of primary production or community's shift due to the reduced light at depth.

Investigating the potential effects of climate change involves not only surface processes, but also deep systems at the
large and local scales, especially where multiple local changes (i.e. wind turbine deployments changing levels of mixing)
repeated over large spatial areas (i.e. the North Sea) are likely to have an effect at different scales (van der Molen et al.,
2014; De Dominicis et al., 2018). Long-term effects of variations in deep mixing processes appear essential to assess
shelf seas at a regional scale, leading to identifying key indicators, or sensitive links, of subsurface highly productive
patches at a fine scale. The physical processes delineating the vertical distribution of density therefore represented a
valuable tool in identifying possible biases or underestimations of Chl-a contents in shelf waters.

### 5.   Conclusion

Chl-a vertical distribution (here classified as shapes) gives important information about the state of development of the
phytoplankton community and their reliance on nutrient gradients that are likely to be associated with mixed and stratified
layers. The upper and deep mixing processes can have very different influences on the Chl-a vertical distribution, dictating
the concentration at subsurface patches that can distribute close to, above, or below DMC.

The association of phytoplankton with AMLD has been largely described at large spatial scales within oceanic habitats.
This study shows there is a very weak linkage between AMLD and DMC at a very high resolution (vertical samples at 1
m distances) compared to HPDs' indicators or BMLD, which has led us to hypothesize that, at fine spatial scales, in
shallow shelf seas, there is a stricter association of summertime subsurface patches of Chl-a with the bottom half of the
pycnocline. Therefore bottom mixing processes (e.g. tidal cycles) may play a role in regulating summertime subsurface
primary production in shelf waters. Considering the described associations of subsurface Chl-a with BMLD provided by
this study, it is evident how this new level of understanding can play a role in the assessment of productivity, since the
bottom mixing processes may be more (or equally) relevant than the surface process in determining a shift of primary
production at a local (due to e.g. the increase of mixing downstream a wind turbine deployment) or large scales (e.g. due
to climate change). This association therefore advocates the  investigation of the effect of anomaly-inducing processes
occurring at and below the pycnocline (e.g. bottom sea temperature, bottom salinity, turbulence and physical processes
at the BMLD), which are likely to influence primary production and the whole ecosystem dynamics within shelf seas
(Trifonova et al., 2021). The  new understanding of mechanisms affecting primary production at fine scales may be very
important to investigate as we are moving rapidly towards the deployment of thousands of wind turbine foundations and
100s of GW of wind energy extraction from worldwide shallow seas (Gielen et al., 2019).  Hence, BMLD is proposed as




an ecological relevant variable for further oceanographic investigations in shelf waters, and the proposed approach is a

valuable tool to extrapolate this variable from *in situ* vertical samples.

**Appendix A**

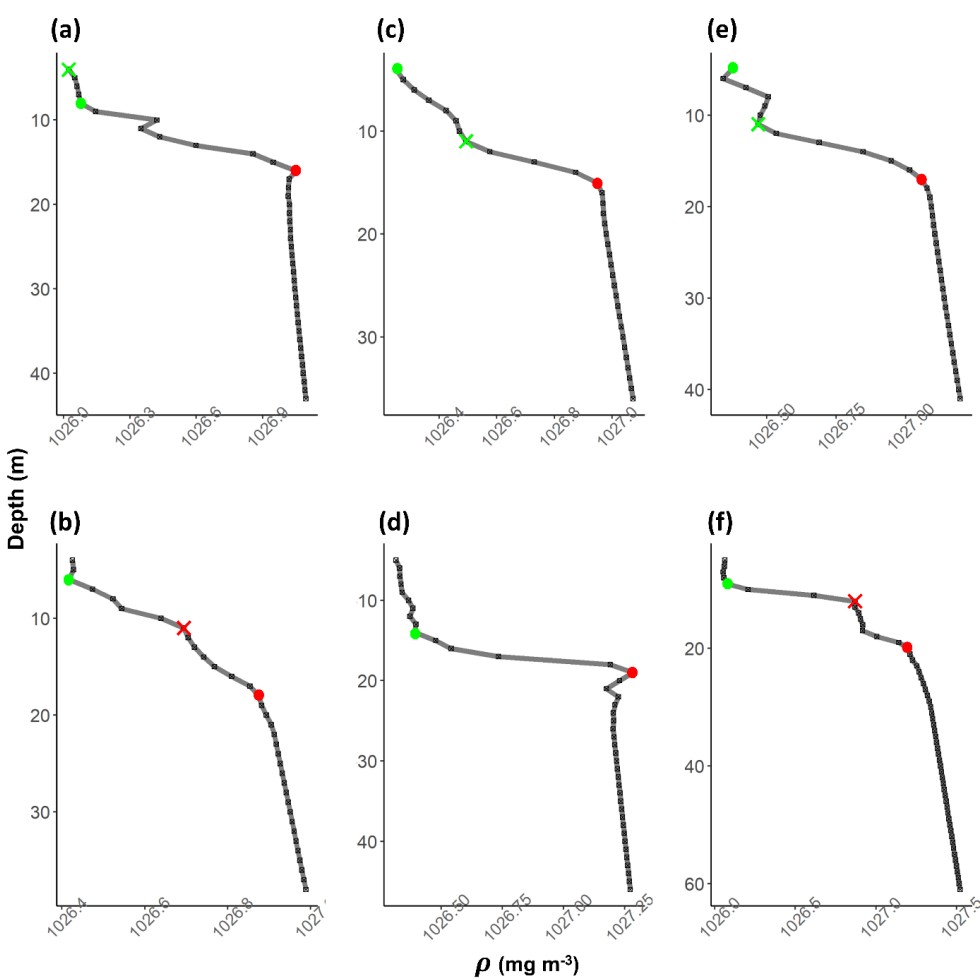

*Figure A1: Examples of density profiles (grey line) (a-f). The black squares are the values at 1 m resolution. Red dots refer to BMLD, green dots to AMLD. Crosses refer to misidentified AMLD (in green) and BMLD (in red) that needed to*

*be manually corrected.*



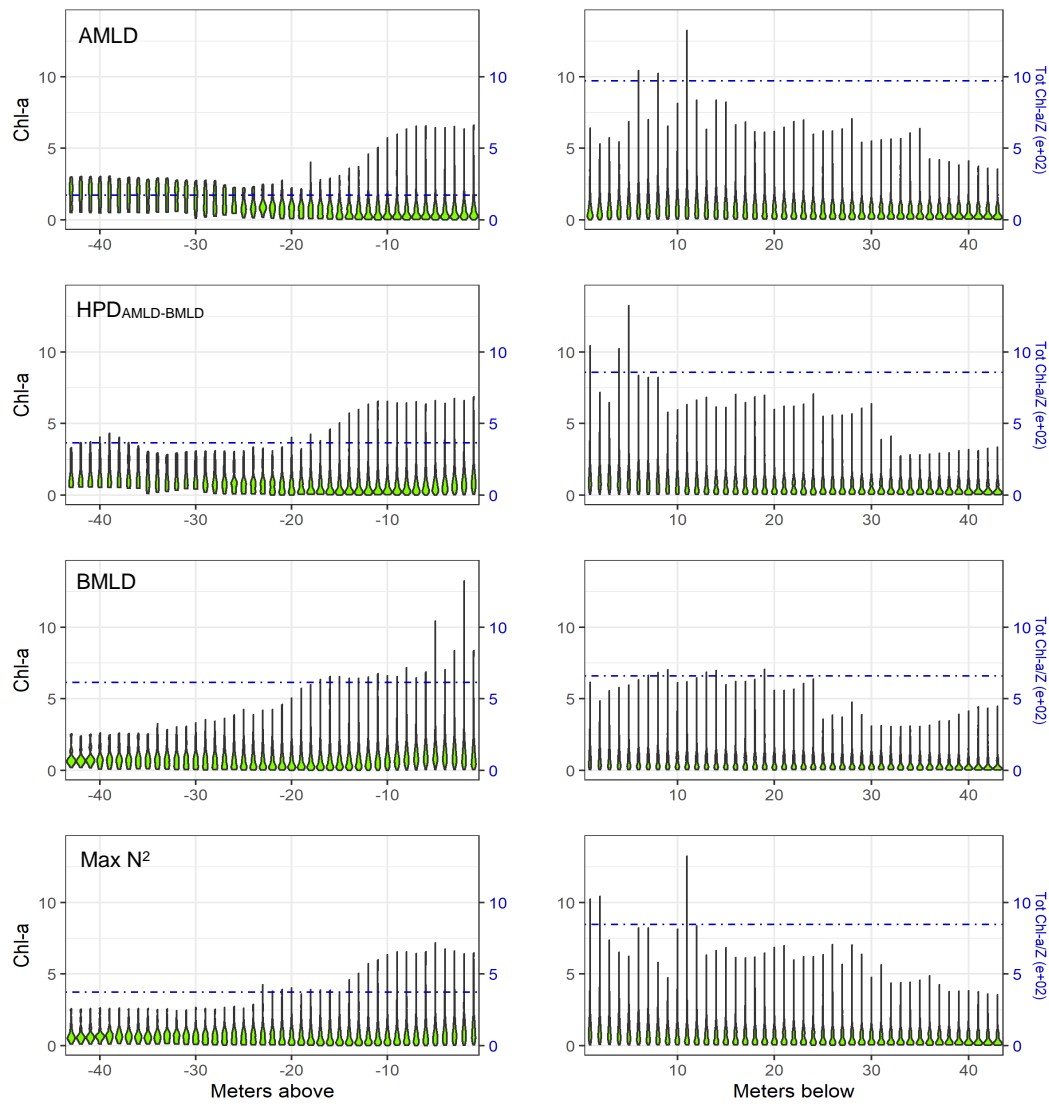

*Figure A2: Violin plot of the amount of Chl-a (mg) at each meter above and below the four density layers (AMLD,*
*HPD$_{AMLD-BMLD}$, BMLD and Max N$^2$) from the whole dataset. The dot-dashed blue lines represent the depth-integrated*
*Chl-a measured as the total amount of Chl-a (mg) divided by the number of depths (z) within each portion of the water*
*column (meters above and meters below DLs) (values are reported in Table 2).*





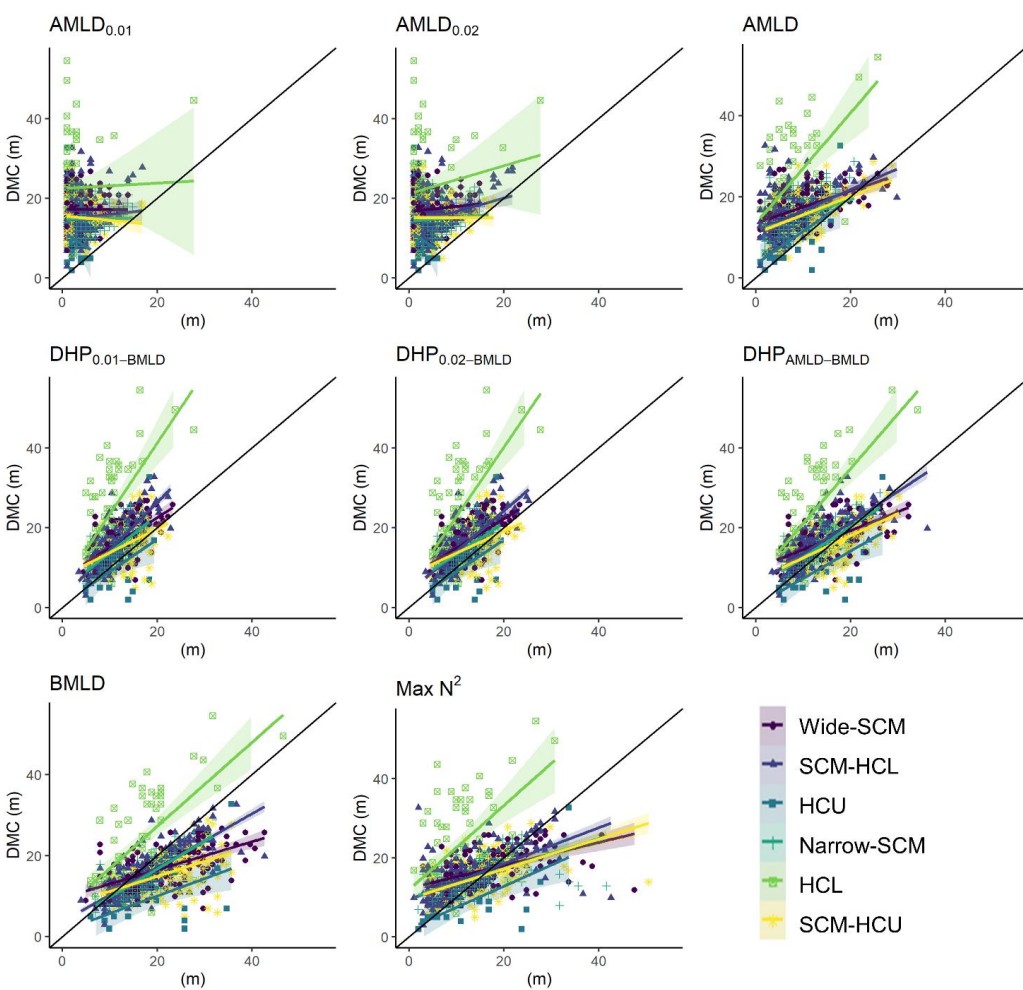

*Figure A3: Plots of DMCs against the eight investigated density layers, whose observations are coloured by Chl-a vertical shape. Coloured lines refer to the empirical linear regression (DMC ~ DL), while the black solid line is the one-to-one fitting-line (DMC = DL).*






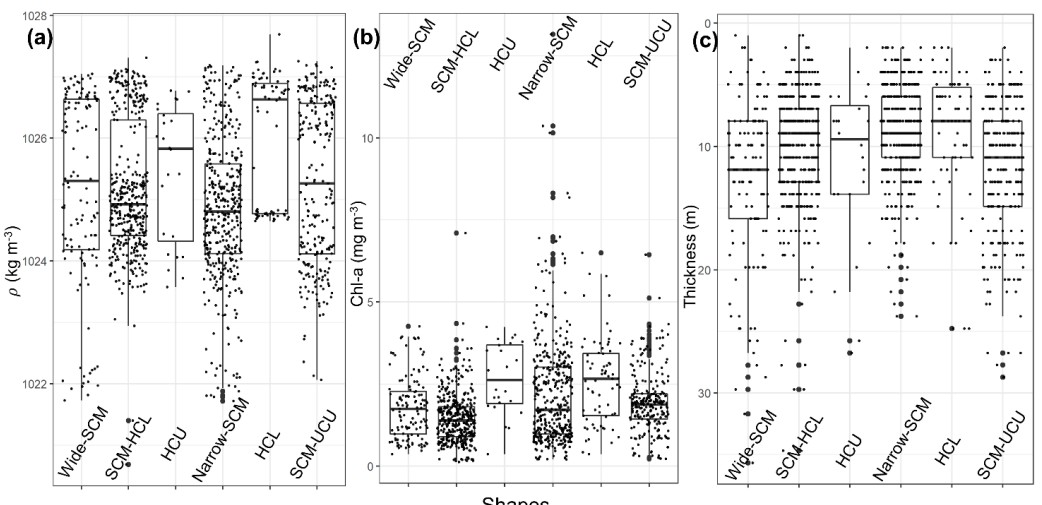

*Figure A4: Boxplots of (a) density at DMCs, (b) Chl-a at DMCs, and (c) the thickness of pycnoclines (measured as the*
735            *difference between AMLD and BMLD) for each Chl-a shape.*

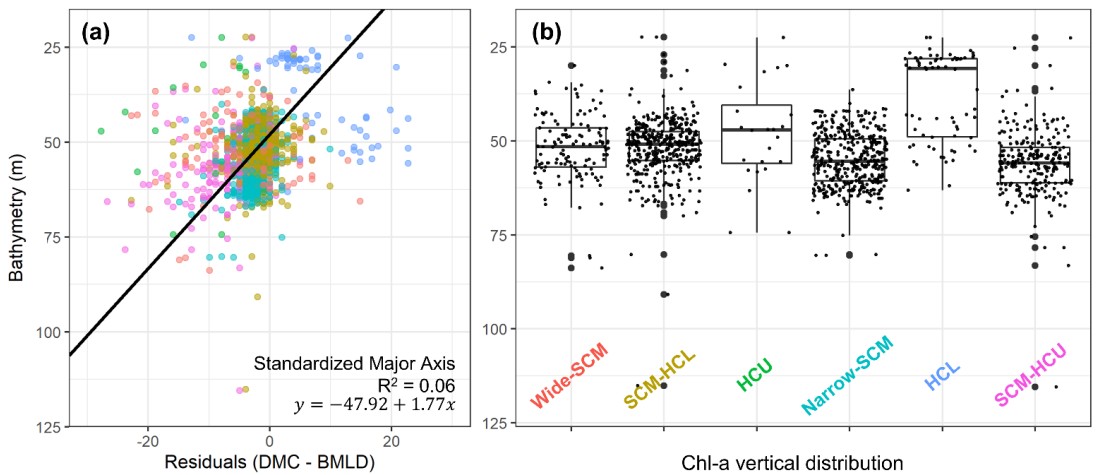

*Figure A5: (a) scatterplot of the residuals measured as the difference between DMC and BMLD (one-to-one fitting-line,*
*DMC=BMLD), against the bathymetry at which each profile was sampled. (b) the solid black line reports a*
*Standardized Major Axis analysis. Colours refer to Chl-a shape, whose ranges of bathymetry.*


*Tables A1-A7:* Statistical parameters and percentages of the observations categorized by Chl-a vertical shape exhibiting
DMCs above (>), at the same depth (=), or below (<) the $AMLD_{0.01}$ (Table A1), $AMLD_{0.02}$ (Table A2), AMLD (Table
A3), $HPD_{0.01\text{-}BMLD}$ (Table A4), $HPD_{0.02\text{-}BMLD}$ (Table A5), $HPD_{AMLD\text{-}BMLD}$ (Table A6), and Max $N^2$ (Table A7).

*Table A1*

| $DL = AMLD_{0.01}$ |
|---|



| Chl-a shape | $\rho_S$ | $\alpha$ | $\beta$ | $R_0^2$ | $R_{em}^2$ | DMC > DL | DMC = DL | DMC < DL |
|---|---|---|---|---|---|---|---|---|
| Wide-SCM | 0.00 | 21339.29 | -5546.12 | 0.35 | 0.00 | 100 | 0.00 | 0.00 |
| SCM-HCL | 0.07 | -83.35 | 24.55 | 0.41 | 0.00 | 100 | 0.00 | 0.00 |
| HCU | -0.18 | 68.85 | -23.09 | 0.27 | 0.03 | 79.17 | 16.67 | 4.17 |
| Narrow-SCM | 0.02 | -127.00 | 28.14 | 0.51 | 0.00 | 100 | 0.00 | 0.00 |
| HCL | 0.02 | -436.78 | 129.32 | 0.22 | 0.00 | 100 | 0.00 | 0.00 |
| SCM-HCU | -0.07 | 68.89 | -13.77 | 0.38 | 0.01 | 99.59 | 0.41 | 0.00 |


*Table A2*

| DL = AMLD$_{0.02}$ | | | | | | | | |
|---|---|---|---|---|---|---|---|---|
| Chl-a shape | $\rho_S$ | $\alpha$ | $\beta$ | $R_0^2$ | $R_{em}^2$ | DMC > DL | DMC = DL | DMC < DL |
| Wide-SCM | 0.08 | -32.36 | 11.67 | 0.38 | 0.01 | 100 | 0.00 | 0.00 |
| SCM-HCL | 0.24 | -0.27 | 3.12 | 0.49 | 0.06 | 100 | 0.00 | 0.00 |
| HCU | -0.18 | 69.85 | -22.75 | 0.28 | 0.03 | 79.17 | 16.67 | 4.17 |
| Narrow-SCM | 0.09 | 6.51 | 1.30 | 0.61 | 0.01 | 100 | 0.00 | 0.00 |
| HCL | 0.13 | -49.80 | 16.96 | 0.27 | 0.02 | 100 | 0.00 | 0.00 |
| SCM-HCU | 0.00 | 776.92 | -158.95 | 0.45 | 0.00 | 99.18 | 0.00 | 0.82 |

*Table A3*

| DL = AMLD | | | | | | | | |
|---|---|---|---|---|---|---|---|---|
| Chl-a shape | $\rho_S$ | $\alpha$ | $\beta$ | $R_0^2$ | $R_{em}^2$ | DMC > DL | DMC = DL | DMC < DL |
| Wide-SCM | 0.48 | 10.88 | 0.69 | 0.70 | 0.23 | 91.20 | 1.60 | 7.20 |
| SCM-HCL | 0.51 | 7.76 | 1.04 | 0.66 | 0.26 | 98.52 | 0.49 | 0.99 |
| HCU | 0.58 | -3.77 | 1.73 | 0.76 | 0.34 | 62.50 | 12.50 | 25.00 |
| Narrow-SCM | 0.41 | 7.40 | 0.88 | 0.77 | 0.17 | 98.76 | 1.24 | 0.00 |
| HCL | 0.55 | -4.23 | 3.97 | 0.47 | 0.31 | 98.57 | 0.00 | 1.43 |
| SCM-HCU | 0.51 | 8.00 | 0.79 | 0.77 | 0.26 | 91.43 | 4.08 | 4.49 |

*Table A4*

| DL = HPD$_{AMLD\ 0.01-BMLD}$ | | | | | | | | |
|---|---|---|---|---|---|---|---|---|
| Chl-a shape | $\rho_S$ | $\alpha$ | $\beta$ | $R_0^2$ | $R_{em}^2$ | DMC > DL | DMC = DL | DMC < DL |
| Wide-SCM | 0.60 | -1.68 | 1.46 | 0.89 | 0.36 | 88.80 | 1.60 | 9.60 |
| SCM-HCL | 0.75 | -4.06 | 1.81 | 0.88 | 0.57 | 95.31 | 1.98 | 2.72 |
| HCU | 0.52 | -14.66 | 2.24 | 0.76 | 0.27 | 41.67 | 4.17 | 54.17 |
| Narrow-SCM | 0.64 | -5.18 | 1.79 | 0.91 | 0.41 | 94.31 | 1.49 | 4.21 |
| HCL | 0.65 | -14.46 | 3.88 | 0.61 | 0.43 | 97.14 | 0.00 | 2.86 |
| SCM-HCU | 0.43 | -9.78 | 2.03 | 0.90 | 0.19 | 78.37 | 2.45 | 19.18 |

*Table A5*

| DL = HPD$_{AMLD\ 0.02-BMLD}$ |
|---|




| Chl-a shape | $\rho_S$ | $\alpha$ | $\beta$ | $R_0^2$ | $R_{em}^2$ | DMC > DL | DMC = DL | DMC < DL |
|---|---|---|---|---|---|---|---|---|
| Wide-SCM | 0.61 | -0.61 | 1.36 | 0.90 | 0.37 | 87.20 | 2.40 | 10.40 |
| SCM-HCL | 0.74 | -1.55 | 1.51 | 0.90 | 0.56 | 94.07 | 2.72 | 3.21 |
| HCU | 0.52 | -14.68 | 2.23 | 0.76 | 0.27 | 41.67 | 4.17 | 54.17 |
| Narrow-SCM | 0.61 | -3.03 | 1.51 | 0.93 | 0.37 | 86.88 | 5.45 | 7.67 |
| HCL | 0.67 | -12.50 | 3.54 | 0.63 | 0.45 | 97.14 | 0.00 | 2.86 |
| SCM-HCU | 0.43 | -7.01 | 1.74 | 0.91 | 0.18 | 73.88 | 4.49 | 21.63 |

*Table A6*

| DL = HPD$_{AMLD-BMLD}$ | | | | | | | | |
|---|---|---|---|---|---|---|---|---|
| Chl-a shape | $\rho_S$ | $\alpha$ | $\beta$ | $R_0^2$ | $R_{em}^2$ | DMC > DL | DMC = DL | DMC < DL |
| Wide-SCM | 0.60 | 6.65 | 0.68 | 0.91 | 0.36 | 66.40 | 1.60 | 32.00 |
| SCM-HCL | 0.74 | 2.20 | 1.07 | 0.92 | 0.56 | 87.65 | 3.95 | 8.40 |
| HCU | 0.62 | -6.13 | 1.17 | 0.68 | 0.38 | 20.83 | 4.17 | 75.00 |
| Narrow-SCM | 0.71 | 0.22 | 1.13 | 0.96 | 0.50 | 75.25 | 5.94 | 18.81 |
| HCL | 0.69 | -5.74 | 2.54 | 0.69 | 0.48 | 95.71 | 0.00 | 4.29 |
| SCM-HCU | 0.59 | 1.68 | 0.91 | 0.94 | 0.35 | 56.73 | 6.53 | 36.73 |


*Table A7*

| DL = Max N$^2$ | | | | | | | | |
|---|---|---|---|---|---|---|---|---|
| Chl-a shape | $\rho_S$ | $\alpha$ | $\beta$ | $R_0^2$ | $R_{em}^2$ | DMC > DL | DMC = DL | DMC < DL |
| Wide-SCM | 0.51 | 10.95 | 0.37 | 0.83 | 0.26 | 56.00 | 5.60 | 38.40 |
| SCM-HCL | 0.63 | 7.57 | 0.61 | 0.88 | 0.39 | 73.09 | 13.83 | 13.09 |
| HCU | 0.55 | 4.42 | 0.34 | -0.11 | 0.31 | 16.67 | 16.67 | 66.67 |
| Narrow-SCM | 0.55 | 7.82 | 0.52 | 0.92 | 0.30 | 64.85 | 16.58 | 18.56 |
| HCL | 0.56 | -5.84 | 2.82 | 0.62 | 0.31 | 95.71 | 1.43 | 2.86 |
| SCM-HCU | 0.55 | 7.52 | 0.52 | 0.89 | 0.30 | 52.24 | 15.10 | 32.65 |

*Table A8: Wilcoxon test between the density at DMCs in HCL shape and all the other Chl-a shapes. In bold the Chl-a shapes having density at DMCs significantly different from HCL profiles.*

| *Shape* vs HCL | W | *p* |
|---|---|---|
| Wide-SCM | 4375 | 0.289 |
| SCM-HCL | 15624 | 0.062 |
| HCU | 1075 | 0.126 |
| Narrow-SCM | 19592 | **0.023** |
| SCM-HCU | 11824 | **0.000** |


## Author contribution

Arianna Zampollo contributed to the conceptualization of the study, formal analyses, methodology on AMLD and
BMLD, writing of the original draft, and software use; Thomas Cornulier contributed to the conceptualization and

 

supervision of the statistical method, writing of the original draft, methodology and visualization of the results; Rory O'Hara Murray contributed on the data curation, writing of the original draft, supervision, visualization and validation; Jacqueline F. Tweddle contributed to the conceptualization and the supervision of the study; James Dunning contributed to the methodology of the AMLD and BMLD algorithm; Beth Scott contributed to the conceptualization of the analyses, writing of the original draft, supervision, funding acquisition, resources and data curation.

**Code availability**

The code for the AMLD and BMLD algorithm are available upon request to zampolloarianna@gmail.com

**Data availability**

Data are available upon request and agreement with the co-authors.

**Competing interests**

The authors declare that they have no conflict of interest.

**Acknowledgment**

The authors thank the founding MarCRF, the Marine Collaboration Research Forum jointly sponsored by the University of Aberdeen and Marine Scotland Science, and Marine Scotland Science to provide a portion of the data.

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
