# Peer review of "A proxy of subsurface Chlorophyll-a in shelf waters: use of density profiles and the below mixed layer depth (BMLD)"

_EGUsphere, 2022_

## Author Response (AR1)

**Dear reviewer,**

We are very thankful for your time and your comments on the paper. According to all the reviewers, we identified some common issues that came across, and we have planned to improve the manuscript following all your advice.

The main points we worked on are: i) better defining the scope of the paper by deleting the Chl-a shapes from the analyses, ii) simplifying the methods, and iii) providing the code to let users trying with the proposed algorithm.

Below, we describe the main changes we introduced into the paper to address the above points.

The scope of the paper has been clarified by focusing on the BMLD (base of the pycnocline) and its use as a proxy for the depth of maximum Chl-a (DMC) in shelf waters. The paper was packed with many details regarding the co-occurrence at the same depth of any density layer (that we renamed "level") (e.g. AMLD, BMLD, DHP and Max N2) and DMC (that we renamed CMd following a comment of reviewer 3). The paper was reporting first the comparison for all the profiles together (section 3.2) and then the comparison for each Chl-a shape (section 3.3). However, the length of the paper and the amount of information were creating confusion among all the reviewers, who struggled to identify the main scope of the paper and often focused mainly on issues referred to Chl-a shapes. On the contrary, we have written this paper to promote a different point of view in investigating subsurface Chl-a by using density profiles. Hence, the main aim of the paper is to highlight the BMLD as a useful tool to predict and investigate CMd in shelf waters. The vertical distribution of CMd nearby BMLDs suggests that this variable has an ecological relevance when we investigate the vertical distribution of Chl-a subsurface patches, and we suggest its use in further research (enlarging these applications in the Discussion). However, this point did not come across easily, and we decided to delete all the analyses related to Chl-a shapes to focus mainly on the use of the BMLD and its potential. The following paragraphs will be deleted: 2.2 in the methods will not include Chl-a shape identification, 3.3 in the results, 4.1 and 4.2 in the discussion. However, understanding the physical processes underpinning the vertical distribution of each Chl-a shape is an open question, and the presented results showed how each shape exhibits a different association of CMd with the pycnocline. We are interested in detailing this question in another paper, to avoid hiding the main scopes of this paper, which are i) proposing a method to extrapolate the base of the pycnocline from density profiles and ii) evaluating its association with the vertical distribution of Chl-a (regardless the Chl-a shape).

The second and third points ("simplifying the methods" and "providing the code to let users trying with the proposed algorithm") are ensuring that the reader fully understands the method and its potentialities. For this reason, we reduced the number of details regarding the algorithm in paragraph 2.4 and we focused on the requirements, limitations, and circumstances in which the method can be used. We integrated the repetitions in the results into the methods together with figure A1 (now figure 5). Moreover, we uploaded the code of the function on GitHub (https://github.com/azampollo/BMLD), where an example is also provided. The details regarding the structure of the function are reported in the supplementary material to allow people to replicate, improve and use the code. Therefore, the diagram and part of the methods are moved to supplementary materials.

The removal of Chl-a shapes from the paper changed the discussion section, which has been reduced and focused on describing the relationship between density and Chl-a profiles. We reviewed the physical variables that are playing a role in the definition of BMLD and AMLD, and their association with the vertical distribution of maximum Chl-a in the water column.

**OVERALL CHANGES**

Heading of Section 1.3 was changed from "A new way forward: the base of the pycnocline (BMLD) as an ecological proxy of the vertical distribution of maxima Chl-a (DMC) in shelf waters" to "A new way forward: the base of the pycnocline (BMLD) as a proxy for maxima Chl-a in shelf waters"

Since section 2.2 "Subsurface Chlorophyll-a parameters" was describing the selection of DMC using a portion of the algorithm proposed in this paper, we moved this section after the section describing AMLD and BMLD detection (now 2.2) "AMLD and BMLD detection". This structure allows the reader to have all the information to processes the method used to detect DMC. Hence, Section 2.4 is now describing the chl-a parameters. The sequence was changed from "2.1 Physical and biological oceanographic samples, 2.1.1 Standardized vertical sampling for density and Chl-a, 2.2 Subsurface Chlorophyll-a parameters, 2.3 Common methods identifying Density Layers (DLs), 2.4 AMLD and BMLD detection, 2.5 Evaluating the association between density levels and subsurface Chl-a" to "2.1 Physical and biological oceanographic samples, 2.1.1 Standardized density profiles, 2.2 AMLD and BMLD detection, 2.3 Common methods identifying Density Layers (DLs), 2.4 Subsurface Chlorophyll-a parameters, 2.5 Evaluating the association of density levels with subsurface Chl-a".

**#REVIEWER 1**

Line 38: "Climate change is introducing...." You could also mention the increasing recognition of possible changes associated with large-scale roll-out of renewable energy in deep shelf seas (e.g. Dorrel et al., 2022: <a href="https://www.frontiersin.org/articles/10.3389/fmars.2022.830927/full">https://www.frontiersin.org/articles/10.3389/fmars.2022.830927/full</a>).

Thank you very much for pointing this out. We changed the sentence that was referring only to climate change, and we added man-made structure as a source of variations in the mixing/stratification balance.

Line 58: "...where the stratification is maintained by tidal cycles mixing the water column through horizontal circulation..." I think this needs rewording. Stratification is not maintained by tidal mixing – the existence and strength of stratification are controlled by a balance between mixing processes (which in NW European shelf seas are generally dominated by tidal mixing) and the source(s) of buoyancy (surface heating and estuarine inputs of low salinity water).

We agreed it was not fully explained and we changed the whole sentence following your comment: "The stratification is generally controlled by mixing processes (tidal mixing and surface wind stress) and sources of buoyancy (surface heating and estuarine inputs of low salinity), whose balance allow primary producers to grow in favourable light and nutrient conditions within the pycnocline. In the North Sea, mixing processes are mostly regulated by strong tidal currents (Glorioso and Simpson, 1994; Loder et al., 1992; Sharples et al., 2006, 2001; Simpson et al., 1980; Zhao et al., 2019b), especially in prolonged stratified conditions, when upward fluxes represent the only source of nutrients intake within the pycnocline."

**Overall, I get a little confused by the term "deep mixing processes". Do you mean mixing at the pycnocline or mixing near the seabed?**

We intend processes occurring below the surface, likely between the pycnocline to the seabed. This has been clarified in the whole paper.

Line 62: a general statement about ocean productivity and climate change should probably also reference something like Steinacher et al., Biogeosciences, 2010. Clarify that the canonical view is that at low and

**temperate latitudes in the open ocean productivity will decrease because of strengthening stratification inhibiting vertical mixing of nutrients.**

We realised that the sentence was referring (again) only to climate change, while we want to refer to any source of disturbance to physics. Hence, it has been changed as follow: However, despite the clear linkage between SCMLs and tidal mixing in shelf seas, variations on productivity have been largely conducted at oceanic sites by investigating the mixing processes above the pycnocline (within the upper mixed layer) (Somavilla et al., 2017; Steinacher et al., 2010), omitting the effects of processes close to the seabed, e.g. variations of mixing processes below the pycnocline.

**Line 74: What is meant by the nutricline exhibiting positive correlations with MLD? What aspects of the nutricline? The depth, the strength?**

**The depth. This information is now added.**

Line 109: Is there a particular reason for the choice of 120 metres as the deepest? Is it simply forced by the data available, or do you have a different reason?

**It is forced by the data available. We deleted the information since it may rise doubts as yours, and it is described in section 2.1.**

Line 118: What does "426 profiles" mean in the context of a mix of towed and vertical-profiling CTD data? Are the individual undulations of the towed systems each counted as a single profile? Is it clearer later in the paragraph – so maybe the full 1273 profiles needs noting here?

We moved the total amount of profiles at the beginning of the paragraph. We also specified the that 426 profiles were collected using the vertical CTD, and that 847 profiles were obtained from the undulating CTD with the sentence ". The continuous profiles obtained from undulating CTD were converted into 847 single profiles of the water columns."

Line 136: "samples' distance" I think should be "sample vertical resolution".

**Changed.**

Section 2.2.1: Why was a GAM/spline used instead of a simple spline (or an even simpler moving average)? Some justification/explanation of this choice would be useful Also, a couple of example profiles in a Fig would help – e.g. one profile where the GAM worked well and another where the visual fixing was required.

I did not know the simple moving average method, and I have read it measure the average of a specified amount of data. I think it would have been a valuable method to fit and smooth the density profiles having a large number of observations, while it may have been limited for those profiles having a few numbers of observations throughout the water column. Here, GAMs were used since it is the method I know better among the smoothing tools, and its application was successful in most of the dataset. Hence, I decided to keep on with this method considering the large number of analyses that the paper was requiring. Hope it is still a valuable tool for the paper. I followed your advice and I added a figure in the Appendix A with two profiles: one where GAM worked and one where the visual fixing was required.

**Line 194: not strictly "density gradient" – the values you state are densities.**

We refer to density threshold and hence we fixed the text (kg m-3).

In two of the Methods sections (2.3 - 2.4) I had to work inordinately hard to see what was going on. I think these sections could be clarified with some better ordering. For instance, AMLD is talked about in section 2.3, but the full description of what it is does not occur until 2.4. There is a raft full of HPDs that pops up line 195-200, but it is unclear what they all mean. If you find yourself having to refer to a section further on in the paper (e.g. line 198 you refer to section 2.4 for the explanation of adjusted AMLD) then you need to rethink how you are structuring the material. You need a clear, logical progression of explanations that does not leapfrog – this is really important, as the reader needs to keep track of a large number of different acronyms and their meanings.

Thank you for suggesting to change the methods. We agreed it was not easy to read, and hence we changed the sequence of sections. The sequence was changed from "2.1 Physical and biological oceanographic samples, 2.1.1 Standardized vertical sampling for density and Chl-a, 2.2 Subsurface Chlorophyll-a parameters, 2.3 Common methods identifying Density Layers (DLs), 2.4 AMLD and BMLD detection, 2.5 Evaluating the association between density levels and subsurface Chl-a" into "2.1 Physical and biological oceanographic samples, 2.1.1 Standardized vertical sampling for density and Chl-a, 2.2 AMLD and BMLD detection, 2.3 Common methods identifying Density Levels (DLs), 2.4 Subsurface Chlorophyll-a parameters, 2.5 Evaluating the association of density levels with subsurface Chl-a".

Line 213: "transient" – do you mean "transition"? Unclear what you are trying to say.

We meant transitional. Transient changed into transitional.

Line 215: delta-rho is a density difference, not a density gradient. This occurs a few times.

Delta-rho was used to refer to a density difference and hence we used the right specification in the whole paper.

Lines 216 – 227: It is really hard to understand what is meant here (partially, but not wholly, because when you say "this paper" I cannot work out if you mean your paper or the Chu & Fan paper cited in the previous sentence). Clarification needed.

We changed the beginning of each sentence for lines 216-227, with "the above assumptions", "such density conditions" and we started a new paragraph after citing Chu&Fun paper. We used "this paper" to refer our paper, and hence we changed the sentence into "In the proposed algorithm, the detection of AMLD.."

Around this stage I just got very confused with the methods. They appear rather complicated and dense, and I found them difficult to follow. To me this difficulty began to detract from what I thought the paper was aiming to demonstrate. Perhaps consider a Supplementary Material section to deal with the details of the methods (though they would still need to be clarified) and focus the main paper on the results and implications?

We agree with your comment, and we decided to move this paragraph to supplementary materials. Hence, the section 2.4 describes now the definition, shortly the method, and circumstances in which the method can be used. The supplementary materials include now the description of the algorithm, and link to GitLab repository where the function can be downloaded and used. A small description of its use, and an example is reported in GitHub.

Section 3.1 starts by repeating a lot of the methods. No real results appear until 3.2 and Fig. 4.

This section has been moved to methods in the new section 2.2.

**#REVIEWER 2**

I.14 Abstract and general. The definition/selection of 8 'density layers' instead of other number is not sufficiently justified. These are levels (discrete depths) instead of layers.

All DL were changed from "layers" into density "levels".

I.36. (also I.57). Specific for shelf seas with strong tides. The authors should notice that many shelf seas have small o no tides.

Thank you for pointing this out. We specified that we refer to the North Sea and that it is characterized by strong tidal mixing.

I.41. Bryden et. al 2005 paper here is not adequate. Scale is too broad and main outcomes are superseded by further results of the rapid array and others.

We changed Bryden et al. 2005 with Orihuela-Pinto et al., 2022 and Bonaduce et al. 2019.

1.89-90. There are no standard methods to MLD identification neither in shelf nor oceanic waters.

Sentence changed.

I.91. BMLD as an "indicator" of the vertical... Indicator or proxy?

Thank you for pointing this out. We changed "indicator" with "proxy".

I.96 and others. BMLD is indistinctly referred to as 'bottom mixed layer depth' and 'below mld". Should address this mismatch.

All the "bottom mixed layer depth" are changed into "below mixed layer depth" to be consistent.

I.101 'this new level of understanding' sounds a bit presumptuous, maybe just this new algorithm.

"This new level of understanding" changed with "This approach" since we are talking not only about the method.

I.111. 'Fig.2'. It is normally requested to cite figures in order, please check.

We deleted this reference, together with the figure – moved to supp. materials.

I.123. 'standard MSS editing procedure' requires a reference.

Sentence removed after comment rev. 3.

1.132 and others. Not necessary to specify used functions of TEOS-10, this is too much detail.

Sentence removed from "In situ conductivity were converted.." which included TEOS-10.

I.135 et.seq. (section 2.1.1). General, I guess the authors are using Chlorophyll fluorescence profiles (from a fluorometer) which is not the same as Chlorophyll-a. Should clarify.

Data were calibrated for Chlorophyll-a from Marine Scotland Science. Hence, we specified always chlorophyll-a instead of fluorescence.

I.138. Understand that smoothing/resampling refers only to undulator.

Correct. We specified it by adding ".. 0.5 to 1 m from undulating CTD".

I.149 'The analyses were run in R v3.6.3...' too much detailed. Again in I.204 etc.

We prefer to refer to the functions and software that were used to allow a perfect replication of the methods. In particular, the new versions of R may not load correctly old packages, and hence functions that are used today may not be used in the future R versions. Therefore, we agreed to specify the version of R that we used.

l.161-162. I do not understand sentence 'and three equal sections were used to divide the difference between the minimum and maximum Chl-a values into three equal sections'

This sentence was deleted with Chl-a shapes.

I.175. Fig.2. why HCL (e) is above HCU (f)? I find this confusing.

This figure was deleted with Chl-a shapes.

I.191 One of the first comprehensive classifications of MLD objective methods available is provided by Thomson and Fine, JAOT, 2003, including curve segmentation aforementioned methods.

Thank you very much for suggesting this paper. I remember I came across it at the beginning of my investigations and then I focused on the Chu and Fan method. I added it to Sect. 2.3 "common methods to identify AMLD". The reference was added.

I.235 et.seq. why these ad hoc parameters? 2-delta and 90% of the entire profile.

We added the following section to justify the selections:

I.240-244. I find confusing that computing the tangent of the angle phi causes issues but computing the angle does not.

Because the tangent of an angle returns positive values for angles between 0 and 90, and negative values from 90 to 180. Moreover, the tangent of the angle increases from 0 to 90 and decreases from 90 to 180. Hence, selecting the maximum angle between ascending positive and descending negative tangent values is not intuitive. Instead of adding a condition for positive and negative tangent values, we decided to calculate the angle, which is also more consistent with the method. Moreover, calculating the angle is more straightforward if someone is interested in analysing the magnitude of the density gradient at each meter depth.

**I.299 again density layers vs density levels**

Changed.

I.320, Table.2. I miss an explanation for exploring linear regression and 'one-to-one' regression. Should intercept of regression be forced to cross zero for any reason?

We decided to test the simplest hypothesis where the DMC and the density level are at the same depth in the water column. Hence, they locate along a one-to-one linear regression. A forced linear regression with intercept equal to 0 and a linear regression without any limitation are tested. The intercept equal to 0 and a

slope equal to 1 refer to the one-to-one linear regression. If the intercept is 1, and slope is 1, the depths of DMC (CMd) and DL are lagged by 1(e.g. CMd= 1m, DL=2m).

Section 3.3.1. I find too many numeric details and data in the text, should be embodied in tables or figs. Same issue in 4.2.

The sections 3.3.1 and the details referring to Chl-a shapes in 4.2 were deleted with Chl-a shapes.

**#REVIEWER 3**

L. 19 in the abstract: instead of "<=120 m", consider "depth <=120 m".

**Changed.**

L. 42: Maybe "processes" instead of "effects".

We agree the right word is "processes".

Some parts that could be removed or significantly reduced in the introduction, since they seem repetitions or they are not very informative with respect of the MS objectives:

L. 46-47: "The vertical [...] in the marine environment".

Done.

L. 66-68: "The exclusive [...] needs to be investigate further".

**Done.**

**L.82-90.**

We believe that this section should be left since the main scope of the paper is the mixed layer depth (MLD). These lines refer to reviews mentioning other methods, and papers describing different approach to measure the MLD.

L. 95: Are you meaning "the distance" instead of "the depth"?

We intended the depth at which e.g. the pycnocline starts and the upper mixed layer ends. Hence, we changed the sentence from "depth between [...] the surface mixed layer depth" into "depth between the pycnocline and i) the surface mixed layer".

L. 101: "the performance of these two proposed density layers" can be misleading, since it is not evident what a density layer performance mean. Maybe it could be rephrased with "we compared results with other relationships between density layers and Chl-a proposed in literature".

We changed the whole sentence into "The vertical distribution of density and Chl-a profiles are compared and the ecological relevance of BMLD in investigating subsurface Chl-a is detailed."

L. 112: I suggest to consider to replace "identify" with "to identify".

**Changed.**

L. 113 "is evaluated by comparing the vertical distribution of subsurface Chl-a": to clarify the comparison cited in the sentence, I suggest to consider the following rephrasing "is evaluated thanks to comparison of BMLD with the vertical distribution of subsurface Chl-a.

We changed the sentence into "a new method to identify BMLD is proposed, and its potential is evaluated by comparing it with the vertical distribution of subsurface Chl-a".

L. 121: The indication of the years (from 2000 to 2014) can be moved at L. 118, where the time length of measurements is cited firstly.

**Changed.**

Some details about instruments could be probably removed:

L. 122-123: "Temperature and conductivity [...] editing procedure".

**Removed.**

L. 130-133: From "In situ" to the paragraph end.

**Removed.**

L. 141: "predict" is a word that is usually relate to forecast, in this sentence maybe "interpolate" is more appropriate.

We changed "predict" with "interpolate".

L. 144 From "The pre-processing" to the paragraph end: my impression is that this sentence can be shortened removing non-necessary details, or delated.

**We deleted the sentence to avoid repetitions.**

L. 152: Maybe DCM is a more usual way to identify the subsurface (or deep) Chl-a maximum. However, I understand that the authors are aiming at defining an abbreviation for the depth of the Chl-a mximum (that is not strictly DCM, indeed); I suggest to consider CMd (Chl-a maximum depth) to avoid confusion with DCM.

**DMC changed into CMd in the whole paper.**

L. 154: Here Eq. 1 is cited, but It appears three page later. Usually equations are cited more closely to their appearance in a MS. Consider to move the equation and the first time it is cited closer.

We deleted the reference to the equation here since we prefer it later on. We also moved and changed this section to supplementary materials. We changed the sentence with ", by using the adapted Chu and Fan (2011) method described in Sect 2.4 and Supplementary materials".

L. 164-165: Consider to rephrase as follows: "Only 2% of the profiles were excluded from the dataset due to unclear subdivision or very different shapes.

The section was deleted together with Chl-a shapes.

Fig. 2: It would be more consistent to indicate with a letter (a, b, etc.) each sub-plot of the figure (the left plot is not labelled with a letter). In the right plots, the "Depth" arrow should point toward the bottom.

The Figure 2 include only a density profile now. Hence, it has been simplified to one plot.

L. 182: I think that "rectangles" is more suitable than "squares".

The part of the figure with Chl-a shape was deleted.

L. 186: "Among" (capital A) instead of "among".

**Changed.**

L. 200: Maybe "maximum squared buoyancy frequency" instead of "maximum buoyancy frequency squared"?

**Changed.**

L. 210-216 illustrate characteristics of AMLD and BMLD and methods to identify them, however AMLD definition and identification methods are discussed also at lines 189-195. Consider to condensate in a unique paragraph.

**We gathered these sections into 2.2.**

L. 224-227 seem a repetition of the strategy adopted in the MS.

We fixed this repetition by re-writing section 2.2.

L. 228-292: Please, consider to move detail of this method to an Appendix.

Advice accepted. The section 2.3 and 2.4 were changed and a large portion of 2.4 is moved to supplementary materials.

L. 346-360: these lines contain some repetitions of details provided in Methods section. They can be significantly shortened or removed.

This section was integrated to methods 2.2 to avoid repetition and allow the reader to understand what is considered a correct or a wrong identification in our method. We think that the methods are now eased and support the use of the function. To date, the details of the code are described in the supplementary materials and on the GitLab repository https://github.com/azampollo/BMLD.

**L. 392: A bracket is missing after Fig. 4c.**

**Bracket added.**

L. 440: "amount of phytoplankton" is perhaps misleading, since chlorophyll is evaluated here (and not phytoplankton biomass).

**This section was deleted.**

L. 514-516: "demonstrates" seems quite strong in this context. Please, consider "suggest" or "indicate".

"demonstrates" changed into "suggests".

**L. 649-651 and L. 655-660 provide valuable discussion points.**

This section was deleted together with the Chl-a shapes. The discussion has been changed and we hope now is better structured.

---

## Author Response (AR2)

We would like to thank the reviewers for their insightful and helpful comments. Set out below are the general changes we have made and then responses to specific requests /points.

**General Changes**

Following the range of comments made by Reviewer 1, we have changed the title of the paper by deleting "proxy" and specifying BMLD as the mixed layer depth below the pycnocline. Moreover, we set out in the abstract, introduction and discussion that the aim and novelty of the paper is to provide an empirical method for calculating BMLD and provide context and reasoning as to why the BMLD is an ecological important variable and can be used to investigate the variations in the abundance, phenology and vertical distribution of Chl-a caused by deep mixing processes.

Other main changes regard:

- 1. The name of the mixed depth layers: AMLD was replaced with the well-established MLD as suggested by Reviewer 1, and BMLD was better defined as the *mixed layer depth below the pycnocline*.
- 2. The more conventional SCM is now used for Subsurface Chl-a Maxima and
- 3. The more conventional DCM is now used for the Depth of the Chl-a Maxima.
- 4. The function identifying for MLD and BMLD is now provided in GitHub at <a href="https://github.com/azampollo/BMLD">https://github.com/azampollo/BMLD</a>, together with some examples and a brief description of its use.
- 5. We have included the photoacclimation in section 4.2 Vertical distribution of Chl-a and BMLD

**Reviewer 1**

General:

Overall the manuscript has improved by shortening the Methods section and making some progress in clarifying what the paper is all about.

We are pleased that Reviewer 1 has liked all improvements so far

The writing can be quite difficult to decipher at times. Sometimes it feels that it is being made to sound more complicated than it needs to be. I would shorten/split a lot of the sentences and aim for concise clarity. Consider every sentence and ask yourself (1) does this say what I think it says, and (2) does this say something that is important for the paper? You need to have a clear view of who you are writing this for. The paper as it stands is probably well targeted at someone with a high level of data analytic/stats skills who will be familiar with some or most of the methods and terminology. It is not targeted at someone with a more practical/observational oceanographic background. But I suspect you do want to reach that 2nd group, in which case the methods still need to be clarified/distilled. For instance, Fig. 3a, b needs to be used better to provide a step-by-step lead through how AMLD and BMLD are arrived at. You could also consider splitting into shorter paragraphs or sub-sections (e.g. sections 2.2, 2.3 and 2.5 have some long, very detailed paragraphs that are difficult to keep track of).

We see there was more to do in providing clarity – as stated below with many good suggestions that have been taken up. The areas of improvement will be specified in the specific response sections 2.2, 2.3 and 2.5.

**Specifics:**

1. Abstract, line 18: BMLD is introduced here, but appears to be defined as "base of the pycnocline (BMLD)". Presumably BMLD=Bottom Mixed Layer Depth – so that is what you need to define it as when you first mention it. Be clearer about what you mean by determining a "proxy" for the SCM. Do you mean, by looking at a density profile in the absence of any chl data you can say something sensible about where the chl profile would be? Then explain why this might be useful (you say it is essential to investigate the impacts of physical changes, but that sounds rather vague – what examples might help explain your idea here?)

We changed abstract to make it more straight forward. We agreed that "BMLD" was defined differently in the sections of the paper. Hence, we specified in the abstract that "A new algorithm identifying the mixed

layer depths above the pycnocline (AMLD and BMLD) is proposed". BMLD has been then defined together with one of the main aims of the paper: providing the method and the reasons why the BMLD can be used as a good variable to understand the vertical distribution of Chl-a in stratified shelf waters.

2. BMLD, CMd, SCML and probably MLD are poor choices for "keywords". Think of keywords as search terms someone might use that could lead them to your paper. Using acronyms as keywords presumes the searcher already knows exactly what they are looking for.

The keywords were changed to deep mixing, , depth of Chl-a maxima (DCM), subsurface Chl-a maxima (SCM), offshore renewables, primary production

3. Line 31-32: "where the pycnocline acts as a barrier against the mixing of the whole water column and allows cells to buoyance and photosynthesize". I think you mean something like – the pycnocline provides a barrier to mixing between surface and deeper waters, and also a stable habitat for phytoplankton growth in the lower euphotic zone"? I'm not sure what you mean by "buoyance" – you might be invoking some ability of phytoplankton to position themselves within the pycnocline (buoyancy or migration), but such behaviour is not necessary. The key to SCM production is residence time (and acclimation to low light) – buoyancy changes or migration can certainly help that, but they are not critical.

We recognize that the sentence was over-complicated hence we replaced it with "the pycnocline provides a stable habitat for phytoplankton growth in the lower euphotic zone".

**4. Line 33: "the modulation of daily and biweekly strong tidal cycles" and also seasonality?**

We mentioned only daily and biweekly scales because they are related to the ebb and flood, and spring and neap, while the seasonal variation of M2 is relatively limited (5-10%) (Müller et al. 2014, doi: 10.1007/s10236-013-0679-0). However, indeed seasonal variation is still important to include, and we now included "seasonally" in the sentence.

5. Line 35-36: "which can be altered by climate change and man-made infrastructures (Dorrell et al., 2022)." This is a rather sweeping statement that needs a bit more explanation. What aspects of climate change can alter the mixing? The Dorrell reference is fine for man-made structures (you could be more specific and note the impending development of large-scale floating renewable energy structures), but "climate change" needs drawing out more with other supporting references.

For climate change predictions on mixing we have added the references of Holt et al., 2016 and 2018 in the introduction and pick up this now back up in the discussion

6. Line 58: "source of nutrients intake within the pycnocline" sounds awkward. "source of new nutrient supply to the pycnocline"? Also, I think worth specifying "new" as there will be recycling of nutrients within the SCM. This sentence has been moved to L. 33-34 to address the comments from Reviewer 2 since there was a repetition in L. 30-35 and L.53-58. We changed "source of nutrients intake within the pycnocline" into "the main source of new nutrient's supply to the pycnocline".

7. Lines 60-65. I am not sure what point you are trying to make here. You seem to be suggesting that shelf SCM productivity has not been studied much, and mixing below the pycnocline has not been considered (much) in SCM productivity. I can think of work by people such as Hickman, Moore, Holligan, Sharples, Weston, Richardson on the NW European shelf, McManus, Franks, Lucas off W USA (and beyond). I think here is where you are making a statement of what aspect(s) of the SCM you will be focusing on that are novel – and the key to me is the link between the shape of the lower pycnocline (which is a tracer of bottom layer mixing) and the location of the SCM?

Thank you very much to point this out and allow us to explain better our intentions. We did not intend to say that SCMs in shelf waters are little studied, which is quite the opposite especially in the North Sea. Reading the literature, I (Arianna Z) have always found that most of the studies were reporting subsurface concentrations of Chl-a very close to the end of the pycnocline-beginning of the low mixed layer depth, although most of the studies investigating the phenology and abundance of primary production have

considered only MLD (upper mixed layer depth) as an indicator of the possible physical factors influencing these variations.

Hence, the paper aims to promote the use of the BMLD (hereafter intended as the mixed layer depth below the pycnocline) to further assess variations in the abundance, phenology and vertical distribution of Chl-a. A first method is provided to extract the BMLD, which can be surely improved in the future, but offers a new point of view on a very well-known system: the subsurface Chl-a distributes very close to the end of the pycnocline, during summer, in temperate shelf waters. To avoid misunderstanding of our aims, the sentence was changed from "However, despite the clear linkage between SCM and tidal mixing in shelf seas, variations on productivity have been mainly conducted at oceanic sites by investigating the mixing processes above the pycnocline (within the upper mixed layer) (Somavilla et al., 2017; Steinacher et al., 2010), omitting the effects of processes close to the seabed, e.g. variations of mixing processes below the pycnocline. On the other hand, studies on shelf waters suggest variations of the water column due to both surface and deep mixing processes, since the interplay of marine components from surface to seabed are more adjacent than in deep oceanic locations (Durski et al., 2004)" into "However, despite the clear linkage between SCM and deep physical processes in shelf seas, surface mixing processes have been used to investigate the global variations of primary production (Somavilla et al., 2017; Steinacher et al., 2010) making the surface mixed layer depth (MLD) an indicator of variations of Chl-a. The use of MLD is motivated in oceanic sites where the deepest limit of the pycnocline is difficult to draw, while the limits of the pycnocline in shelf waters are more evident due to surface and deep mixings confining the pycnocline in a restricted zone".

The novelty (and aim) of this paper is providing a method and a context to BMLD.

**8. There is a tendency to overload the paper with citations. The first paragraph of section 1.2 has 28 citations, some of which are in lists of 5 - 8. Be more selective – pick the works that are most pertinent, or represent important initial studies.**

We deleted some of the references where the list was unnecessary too long in the whole introduction. Section 1 is still particularly dense and has now 21 citations. We left 5-6 references in the lines 67-71 because we wanted to list all the main papers mentioning the relationship between MLD and Chl-a vertical distribution, bloom events and nutricline depth. Since one of the aims of the paper is proposing BMLD as more informative than MLD, we believe that referring to the main papers that used MLD would set a good background knowledge about MLD and Chl-a relationship.

9. Line 89 (title to 1.3). Again, you appear to define the BMLD as "base of the pycnocline". Then later at line 93 you define it as "below mixed layer depth". I understand what the base of the pycnocline is, but am less sure about how to interpret below the mixed layer depth. I would be very tempted to use MLD and BP: MLD is well-established and does not need redefining to AMLD, and to me BP works better as "base of the pycnocline". That way you are making a clear contrast between the base of the mixed layer and the established concept of the mixed layer depth.

Thank you for being supportive and finding a solution for the acronymous. We agreed to change AMLD to MLD considering its broad use in the literature. On the other hand, we opted to define the BMLD as the *mixed layer depth below the pycnocline*, leaving the order of the letters as the previous version. However, we considered to use BP (base of the pycnocline), but we believe it will deviate it from the concept of being a depth between the pycnocline and a mixed layer. In fact, MLD and BMLD refers to the deepest limit of the upper mixed layer and the first depth of the lower mixed layer (below the pycnocline). The use of BMLD includes in the acronymous the "mixed layer" concept, which is not excluded in BP. Since MLD and BMLD can also be seen as the limits of the pycnocline, the start and end, the top and base of the pycnocline, using BP would associate more with the "top of the pycnocline", which I (Arianna Zampollo) believe should also be defined. Moreover, we believe that defining the base of the pycnocline as the BMLD would increase its exposure. We also considered MLDb, although it appeared too similar to MLD and may lead easily to errors of misspelling.

10. The first paragraph of Methods does not work well. You are trying to summarise things that have yet to be explained, and also repeat/paraphrase the paper's aim which is not necessary here.

**We understand why it is not working and we deleted it.**

**11. Line 143-144: "both transitional layers from a mixed to a stratified vertical region occurring at the beginning and end of the pycnocline." This sounds unclear. Do you mean that they are both the transition regions between mixed waters and the pycnocline?**

Yes, we meant that the locations of MLD and BMLD are the transition regions between mixed waters and the pycnocline. The sentence was changed from "The surface mixed layer depth (AMLD) and the mixed layer depth below the pycnocline (BMLD) are both transitional layers from a mixed to a stratified vertical region occurring at the beginning and end of the pycnocline." into "The upper mixed layer depth (MLD) and the mixed layer depth below the pycnocline (BMLD) are both the transition regions between mixed waters and the pycnocline."

**12. Lines 150-158. This is a confusing paragraph, but I think the point you are making is that both the AMLD and BMLD are defined based on a critical value of the density difference between adjacent data points in the profile? And is this critical density different the same for AMLD and BMLD?**

The selection of MLD and BMLD is not based on a critical value, as the algorithm works regardless for any a priori threshold. The aim of this paragraph was defining how MLD and BMLD are intended in this paper, since using a method without a critical value leads to select a different MLD from those obtained by thresholds' methods (MLD0.1 and MLD0.2). After having introduced the limitations of other common methods, we listed some of the definitions related to MLD and BMLD. Since the paragraph appeared confusing, we simplified it and changed it into "In the proposed algorithm, the detection of MLD does not assume that the upper mixed layer has a density gradient close to zero up to the top of the pycnocline, and it identifies MLD (and BMLD) regardless any *a priori* threshold (Chu and Fan, 2019, 2011; Holte and Talley, 2009). Two approaches, the angle's method from Chu and Fan (2011) and K-mean statistics, are used to analyse the vertical distribution of density ( $\rho$ ) by comparing the observations to each other in the same profile instead of applying an absolute threshold to all profiles. The algorithm distinguishes in the water column three layers having similar density values (the upper mixed layer, pycnocline and lower mixed layer) (Fig. 2) using K-mean statistics. The MLD represents the shallowest depth up to which the difference of density between adjacent points  $\Delta \rho$  is small and similar from the surface. The BMLD is the first depth below the pycnocline from which  $\Delta \rho$  is small and similar down to the seabed. This type of detection based on the density shape allows the identification for unconventional density vertical distribution (Fig. A1 in Appendix A)."

Figure 2 (below) was replaced with the current Figure 2 to support readers visualizing what we intend with upper mixed layer, pycnocline and below mixed layer, and where MLD and BMLD locate.

Density (mg m-3)

**13. Lines 163-185. I think this is the section where I need to understand how AMLD and BMLD are arrived at. But I cannot understand what is happening.**

We acknowledge that the reviewer spent time to understand the method and that this paragraph was quite difficult to decipher. The aim is to summarize and simplify the details reported in the supplementary material, hence this section was extensively re-written and can now be found between lines 143 and 228. Below we gave an overall description of the method and our replies to specific issues for clarity.

The main issue that has been raised is the lack of clarity in the using of V1 (red line) and V2 (blue line). At each measured point (z) in the density profile , V1 is fitted using z and 2 points ( $2\delta$ ) above it, and V2 is fitted using z and 2 points below it. Therefore, a unique V1 and a unique V2 are calculated for each point of the density profile. The angle ( $\phi$ ) resulting from the intersection of the two lines is measured in degrees using equation 1 in Supplementary material. A value of  $\phi$  is hence associated with each point of the density profile. At this point, following Chu and Fan (2011), the maximum angle is chosen as MLD or BMLD. Since the identifications of MLD and BMLD are both based on a ranking of  $\phi$ , the selection of either one or the other requires splitting the observations in the water column to avoid their mis-identification. Therefore, to distinguish MLD from BMLD, the density profile must be split in two sections: Split1 and Split2. If the profile is not split, it would be impossible to distinguish both MLD and BMLD.

Therefore, splitting the profile in "surface" and "deep" sections is necessary to select which observations are used to identify MLD and BMLD. The surface layer goes from the bottom of the pycnocline to the sea surface, and the deepest section goes from half of the pycnocline (which is approximated as the middle point between the minimum and maximum density values) to the deepest recorded point. However, since V1 and V2 are calculated using 3 observations, the half of the pycnocline appeared too close to MLD in very thin pycnoclines (< 5 observations), and led to errors. Hence, Split2 was set to start 2 observations below the half of the pycnocline, while Split1 is just measured using all the observations from BMLD to the sea surface. It is noticeable that the limit of Split 1 is depending on the identification of BMLD, which is solved before the MLD in the algorithm. Following this explanation and the issues raised by the Reviewer, we changed several parts in the paragraph and simplified the algorithm's process.

**Below some additional explanations referred to specific comments from the reviewer:**

"Split1 appears to need knowledge of BMLD before it can be set. I cannot work out how split1 and split2 are used to determine AMLD and BMLD. Is it that split2 is first used to define BMLD, and then BMLD is used to determine split1 which then allows estimation of AMLD?"

Yes, first BMLD is identified using Split2, then BMLD is used to set the limits of Split1 and select the observations used in MLD's identification.

**"But I do not understand how the red and blue lines in Fig 3 are decided upon and the role they play in AMLD and BMLD"**

The intersection between red (V1) and blue (V2) lines return the angle ( $\phi$ ) which is used to identify MLD and BMLD as reported Chu and Fan (2011). However, our high-resolution profiles required a second level of inspection on the MLD and BMLD, which was made with K-mean statistic applied on several candidates (3 for MLD and 5 for BMLD) to select the truest MLD and BMLD.

"I think I can see how two of the lines are set (for AMLD, the blue line is a linear regression on the surface data? For the BMLD the red line is a linear regression on the bottom layer data? Why not use the same colour for these two?)"

This is partially correct, since the blue line in Fig. 3a refers to the regression made with the MLD and 2 observations above it (hence being part of surface data), and the red line in Fig. 3b is the regression made with BMLD and 2 observations below it (hence being part of deep data). The colours refer to the way they are measured: the red line is the regression made out with the investigated depth (z) and 2 observations above it (z-2), the blue line is the regression of z and z+2 (2 observations below z).

"Then does the blue line in Fig 3b allow determination of BMLD, which then allows split1 to be determined and so the red line in Fig. 3a? But what sets the left end of the red line in 3a?"

The blue line and red line contribute to determine MLD in Split1, and to determine BMLD in split2. BMLD contributes to determine split1, which is used to select the observations used in measuring V1 and V2 for MLD's identification.

Also, does this method only work if the data reach very close to the seabed? Split2 seems to be dependent on this. I am also a bit concerned that there were some profiles with density decreasing below the pycnocline – Fig A1d could show a real, temporary overturn in the density profile, but how does the method overcome this? The answer to the question "does this method only work if the data reach very close to the seabed?" is no. This method works well with high-resolution data (1 m), pycnoclines defined by at least 4 observations, and enough observations after the pycnocline. We limited Split2 to  $0.9\Delta\rho$  (90% of the observations from the sea surface to the deepest observations) because the profiles were exhibiting many points below the pycnocline, slowing down the running time of the profile. Moreover, this set up reflect what has been used by Chu and Fan (2011). However, we decided to allow users to choose if they want to consider the whole profile or just 90% of it. We added some details to the GitHub page (https://github.com/azampollo/BMLD/) to let the user change the setting of the abmld.R function.

```
* * *
```

abmld.R is set up to work with the first 90% of the observations from the surface to the seabed (10% of the deepest points are not used). This setting is not ideal if your profiles have BMLD very close to the end (deep portion) of your density profiles. If you want to run the function using all the points of the profile, in abmld.R you have to comment L. 103-104 and uncomment L. 106-107 as shown below:

**USE L. 103-104 IF YOU WANT TO SET THE BOTTOM LIMIT OF SPLIT2 TO EXCLUDE 10% OF THE DEEPEST OBSERVATIONS**

```
**per15 <- nrow(dd)-round((dd$pressure[nrow(dd)]*10)/100)**
**d <- dd[1:per15,]**
**USE L. 106-107 IF YOU WANT TO USE THE WHOLE DENSITY PROFILE**
per15 <- nrow(dd)
d <- dd[1:per15,]</pre>
```

Moreover, we clarified this point in the paper in the section "method to extract MLD and BMLD" with the sentence "The abmld.R function works well with high-resolution data (1 m), pycnoclines defined by more than 5 observations, and the base of the pycnocline occurring within the 90% of the observations from the surface to the deepest point." and "The bottom limit of Split2 was defined at  $z_{0.9\Delta p}$  following Chu and Fan (2011) to reduce the number of observations close to the seabed. However, the analyses can be extended up to the end of the profile by following the instructions reported at the website https://github.com/azampollo/BMLD.".

**14. Lines 193- : Performance of the algorithm seems to require prior knowledge of AMLD and BMLD to then determine of the algorithm got it right. Is there some automated way of assessing the quality of the calculations? Otherwise, you might as well just select AMLD and BMLD manually.**

Thank you for your comment. Since any method can have an error margin, we measured the algorithm's performance by checking the identifications manually. We had to assume some prior knowledge of MLD and BMLD to develop the algorithm and assess its performance. The validation was made with the co-authors and by considering their oceanographic experiences. Obviously, the selection of BMLD, as MLD, can be done manually, but we wanted to provide an automatic method to process large numbers of profiles. Moreover, although many methods are described to identify MLDs, none are reported for BMLD. Hence, the paper aimed to indicate the importance of BMLD and provide a method to extract it easily. Nevertheless, your point is very relevant, and we considered adding the following sentence "The algorithm was validated by manually checking the estimated MLD and BMLD in each profile, which were considered wrongly identified when falling into the pycnocline. Since most of the errors located the mixed layer depths clearly at the centre of the pycnocline having with thin layers of re-stratification (> 4 observations) (Fig. A1 b, c, e, f, Appendix A), the identifications were considered correct when they appeared i) on top of a lower mixed layer (in BMLD) and ii) on top of a large density gradient (pycnocline) separating surface to deep waters (in MLD)". This paragraph clarifies i) that the validation was required to assess the performance of the algorithm, ii) that the validation was manual and iii) what was considered as correct and what was wrong.

**15. Section 2.3. It would help if you first explained what you mean by "density levels" and why they might be/are important here.**

We agreed that the paragraph missed a description for DLs and the reasons behind their selection. Hence, we added the following sentence: "The depths detailing the density structure in the water column are defined here as density levels (DLs). Among the multiple indicators of mixed layers that associate with Chl-a vertical distribution, the ecological relevance of the MLD, the halfway pycnocline depth and the maximum buoyance depth were compared to the proposed algorithm's identifications.

**16. Section 2.4. Not clear how the chl max is determined – you state the same method of angles used in 2.2, but I did not fully grasp that and certainly cannot now see how it is used to determine the chl inflexion point in a way that makes it better then manually doing it.**

Here, as we specified in our reply to comment 14, the manual identification of DCM is a valuable option. However, large dataset can take a lot of time. For example, oceanographic variables from 3D models such as Copernicus dataset, are becoming widely used in spatial distributions models of marine species, and the time of processing several years of daily values can discourage users to adopt very informative variables (such as max Chl-a, MLD and BMLD). Hence, we used the adapted Chu and Fan (2011) method identifying for  $\phi$  angle to automatically pinpoint Max Chl-a (SCM) in the water column. The method is coded into a function named maxChla.R that is available at https://github.com/azampollo/BMLD.

We have re-written this whole section, highlighting that "The angle ( $\phi$ ) were measured at each depth of the Chl-a profile, and the maximum  $\phi$  with the largest Chl-a concentration was selected as DCM". This should clarify that the maximum angle's method was used to identify which depths have a large variation in Chl-a, and the DCM having a large concentration in Chl-a was selected among them. We decided to not give more details about the maximum angle's method because this is already described in Sect 2.2, Supplementary materials, and the github webpage.

17. Section 2.5. The first sentence works well -1 immediately grasped what the aim of this section is. I was less clear one why 3 different linear models were used -1 suspect that each method provides different information, but this was all presented in a fairly dense paragraph. Maybe split them out into their own short paragraphs/subsections so that it is easier for the reader to know what they should be focusing on.

We have added 'All three methods differently assess the level of correlation or prediction'. The aim of using each method is explained in table 2, and in the lines between 274-285.

**18. Line 346 and Table 4: I do not understand where the units of mg m-1 come from. For a depth-integrated chl I would expect mg m-2.**

The correct unit is mg m-3 and it has been corrected. We calculated the depth-integrated Chl-a (mg m-2) and divided this value by the number of observations used to measure the depth-integrated Chl-a. Hence, the standardized depth-integrated (total) Chl-a is the amount of Chl-a in the whole water column above and below DL, and weighted by the number of samplings. Therefore, dividing the total Chl-a (mg m-2) by the number of observations (m) returns mg m-3. We clarified what values in Table 4 represent by changing the legend: "Sum of all depth-integrated Chl-a (mg m-2) standardized by the number of observations above and below the four density layers."

**19. Fig. 5: I do not know what this is showing me (I confess I've not heard of a "violin plot"). I assume they are sowing me the distribution of chl above and below the depth of the maximum across all data. But I do not know exactly how.**

The violin plot is "a hybrid of a box plot and a kernel density plot, which shows peaks in the data" (Joel Carron, Data Scientist at Mode, 2021, link). While the box plot can only show summary statistics, violin plots summarize statistics and the density of the observations. Hence, it returns an idea of how the observations mainly distribute, and (for example) can inform about the ratio of outliers. We decided to use violin plots to represent the amount of Chl-a at each meter depth among all the profiles in order to visualize where most of the Chl-a distribute in the water column. In Figure 5, each violin plot is created with the Chl-a values from all the profiles at any depth. Hence, the violin plot allows visualizing if high concentrations occur at a certain depth beside what is the average value.

**General Discussion point – show specifics**

Overall the Discussion is very hard to grasp. A lot is said, but I find it difficult to pull out what the really important, novel points are from your work. Your results show that on the shelf the SCM tends to be located in the lower pycnocline – which is not surprising. But is a key point that the correlation with the BMLD means that climate predictions of changes in BMLD can tell us about how the SCM might respond? This needs to be much clearer.

We agreed that the link between MLD, BMLD and DCM or primary production was not well explained in the different sections of the discussion. Hence, we simplified some sentences in section 4.1 (paragraph 1), and changed the structure of section 4.2. We also added the photoacclimation effect on section 4.2.

**20. Section 4.1 paragraph 1. So, is your main point here that AMLD probably works better in deep ocean environments where below-pycnocline turbulence is weak?**

We agreed that paragraph 1 was not clear, and we added the sentence "Although MLD are linked to the physical processes setting the vertical distribution of DMCs in deep oceanic environments, all the investigated surface mixed layers' indicators ( $MLD_{0.01}$ ,  $MLD_{0.02}$  and MLD) weakly predicted DCM in the shelf waters investigated in this study."

**21. Lines 403-406. I do not understand what you mean here – particularly, what is meant by "Max N2 would therefore represent a hot spot of nutrients reached by resuspended phytoplankton cells"? Your observations show that the depth of max N2 tends to be above the chl peak, which in every chl peak I have seen in shelf seas will mean that nutrients (nitrate) will be depleted.**

We agreed that the sentence was not entirely correct. We wanted to justify the highest co-occurrence between DCM and Max N2 (13.51%, Table 3) by saying that Max N2 may pick up the layer where Chl-a cumulate due to the less turbidity. However, including the nutrients in this assumption is not correct because their distribution is not influenced only by physical drivers. Therefore, we changed the sentence into "The Max N2 would therefore represent a mild turbulent layer where resuspended phytoplankton cells cumulate, while mixing processes above and/or below Max N2 redistribute phytoplanktonic organisms throughout the water column.".

22. There is potentially a timing issue that needs to be considered. The SCM is a result of weak upward mixing of nutrients, but the slope of the nutricline is also affected by the uptake within the SCM – as summer

progresses the SCM can deepen in the pycnocline, not because the phytoplankton are actively swimming/sinking down but because their uptake of nutrients gradually eats down through the nutricline. All of your data is June/July, so you might not see that – but it is worth considering when comparing with other SCMs.

We are aware of this timing issue and this was the reason behind considering the shape of the vertical distribution of Chl-a in the first version of the paper. We know that subsurface production is strictly related to the nutrients' availability, although data on nutrients are often lacking in large scale analyses. Hence, we want to say in this paper that BMLD can still be an informative variable to understand where phytoplankton distributes in prolonged stratified conditions. We changed the sentences in paragraph 4.3 and added specified that the subsurface production is caused by surface depletion of nutrients after surface blooms: "Prolonged stratified conditions are known to promote subsurface patches of Chl-a (Ross and Sharples 2007; Somavilla et al., 2017) due to the depletion of nutrients at shallow layers after surface blooms. The starvation of nutrients at the surface forces phytoplankton to re-distribute (e.g. Bindoff et al., 2019; Boyd et al., 2015; Schmidt et al., 2020) in deeper nutrient-enriched waters, within the euphotic zone."

**23. Lines 435-436. I do not know what you mean by "invalidated". In fact, I am not sure what I am expected to draw out of this entire paragraph. Towards the end you say "Hence, the location of CMds....", but I cannot see the connection between your statement about the FoF data and the earlier examples in the paragraph. What point are you trying to make?**

Considering the comment made by the chief editor "you should revise your manuscript, particularly the Discussion, with the aim to make clear the new mechanistic insights that your analysis provides, so that your work can be of interest to a wide readership of oceanographers", we decided to clarify the aims in paragraph 4.3 by adding a general introduction "In this section are introduced some of the potential contexts in which BMLD's use would be advantageous. The linkage between the mixed layer depth below the pycnocline and subsurface Chl-a advocates BMLD as a key variable to address the effects of climate changes and man-made structures (e.g. offshore wind farm foundations) on the food resources, and defines BMLD as a potential proxy of subsurface food patches to investigate the vertical and spatial distribution of grazing and predator species" and splitting the paragraph in two sections referring to the potential uses of BMLD in climate change scenarios and offshore renewable infrastructures.

We changed and partially deleted the sentence containing "invalidated".

We hope the intentions in the paragraph are now clearer.

**Reviewer 2**

L. 17: "Out of 1237 observations of the water column exhibiting a pycnocline". "Out of 1237 observations of the water column exhibiting a pycnocline in the North Sea". Sentence is changed.

*L.* 25: A number of Keywords are acronyms, I suggest to consider more explicit keywords. See 'Comment 2' response section to view the new keywords

**L. 92: Are you meaning "the distance between" instead of "the depth between".**

We changed the sentence from "to characterize the depth between the pycnocline and i) the surface mixed layer [...]" into "to identify i) the surface mixed layer [...] and ii) the below mixed layer depth (BMLD) intended as the depth at which the pycnocline ends and the deep mixing develops until the seabed."

L. 53-58: These lines discuss concepts that are partially addressed above (L. 29-33). Please consider to: \* remove "The stratification is generally controlled by [...] nutrient conditions within the pycnocline." Sentence was removed because it repeated L.35-37.

**\* move the following sentence "In the North Sea [...] within the pycnocline" in place of (or integrated with) "The balance between stratification and mixing [...]" (L. 31-33).**

Sentence moved to L. 34-35 and integrated to L.31-35. The sentence was changed from "The balance between stratification and mixing in the water column is determinant for phytoplankton, and, in the North Sea, it fluctuates in time and space by the modulation of daily and biweekly strong tidal cycles (Klymak et al., 2008)." to "The balance between stratification and mixing in the water column is determinant for phytoplankton. In the North Sea, the balance between mixing and stratification fluctuates in time and space by the modulation of daily cycles (Klymak et al., 2008)." to "The balance between stratification and mixing in the water column is determinant for phytoplankton. In the North Sea, the balance between mixing and stratification fluctuates in time and space by the modulation of daily and biweekly strong tidal cycles (Klymak et al., 2008; Loder et al., 1992; Sharples et al., 2006, 2001; Zhao et al., 2019b), which represent the main source of nutrients' input within the pycnocline in prolonged stratified conditions.".

**L. 327-330: This sentence could fit at the beginning of the following paragraph (3.2).**

We agreed that this was repeating the introduction of paragraph 3.2. Hence we integrated L.327-330 into L.337-341. The section changed from "Since hydrodynamic and biological conditions generating resuspension, passive drift, and mortality (i.e. zooplankton grazing in stratified waters) shape Chl-a differently throughout the water column, the amount of Chl-a was measured above and below each density levels regardless the vertical distribution of DCM." to "Although DCM generally reflect the region with the highest concentration of Chl-a throughout the water column, large concentration can still accumulate above or below it. Hydrodynamic and biological conditions generating resuspension, passive drift, and mortality (i.e. zooplankton grazing in stratified waters) can shape Chl-a differently throughout the water column, hence the ecological relevance of the density levels has been investigated in comparison with the vertical distribution of Chl-a".

L. 463: OWFs is introduced here but not used elsewhere in the manuscript. Since several acronyms are already used, the use of OWFs could be avoided. We agreed and deleted "(OWFs)".

---

## Author Response (AR3)

**Dear Prof. Emilio Marañón,**

Thank you for having followed the revision of our manuscript now titled "The bottom mixed layer depth (BMLD) as an indicator of subsurface chlorophyll-a". We received valuable comments from multiple reviewers that clearified and reinforced the aims of the manuscript. Considering the numerous changes that we made on this paper (e.g. exclusion of Chl-a shapes from the first version and definition of BMLD as "bottom mixed layer depth"), we truthfully believe that this final version addresses the ambiguities raised by each of the multiple reviewers and return a knowledgeable manuscript on this topic.

Below we reported the main changes, followed by the responses to specific comments.

**Main changes:**

- We reinforced the definition of BMLD as "bottom mixed layer depth" following the definition from Pingree and Griffiths (1977) and Sharples et al. (2001). The title, manuscript and abbreviations' table (Table 1) were changed.
- We changed "ecological indicator" with "driver" or "indicator".
- We checked whether the definition of BMLD given in this paper is similar to the bottom boundary layer (BBL) as suggested by reviewer 1. We conclude that the two definitions are different with details in the comment's answer below.
- We improved section 1.3 in order to i) state the use of BMLD (which is different from BBL) in understanding physical and biological processes, ii) reinforce the need of a method identifying for BMLD in shelf complex waters, iii) clarify the aims of the paper (return a method to retrieve MLD and BMLD from highly variable density profiles in shelf waters) and iv) justify the comparison among DLs by comparing the vertical distribution of them and Chl-a maxima (DL=DCM). See below.
- In the discussion and conclusions, we added details of our intentions in comparing the vertical distributions of DL and DCM (e.g. BMLD=DCM) regardless any temporal component. The comparisons were made in absence of any variable controlling for the progression of events affecting the physics and biological dynamics of the water column (e.g. vertical Chl-a shape or water column stability). However, the association between any DL and DCM vary depending on the physical and biological conditions of the water column. Hence, we discussed the potential factors involved in the different associations of DCM with MLDs' and HPDs' indicators, Max N2 and BMLD. In general, the MLD is likely to distribute close to DCMs during surface blooms, Max N2 might represent a thin layer where phytoplankton gather (13.51% of the profiles) in a less turbulent region, HPD and BMLD showed the highest correlation to DCMs, while BMLD distributed below DCMs in 78.32% of the profiles. Moreover, we specified that the unexplained variance (scatter points along the 1:1 line) in the linear regressions in Figure 4 is most likely related to the different conditions of the water column, such as the vertical distribution of Chl-a (shapes), nutrients profiles, stability of the water column (transition from either stratified to mixed condition or vice versa), tidal phase, grazing factors, phytoplankton dynamics (e.g. cell's light history, species composition and competition). We suggested that further investigations should be carried out including the factors mentioned above.

- We changed and summarized the conclusions according to the comments.

**#Reviewer 1**

General comment

The Authors made a further effort to include all the suggestions and comments in the manuscript. Some technical aspects have been clarified, while keeping the essential methods and results in the revised manuscript. Discussion has been improved focusing more on the main objectives of the manuscript.

Some minor correction that can further improve the manuscript are listed hereafter:

Consider to do not insert unnecessary acronyms in the abstract, e.g. SCM and MaxN2 are not further used in the abstract itself. On the other hand, MLD (L. 20) is not explained in its first occurrence but then used at the abstract end.

We deleted all the acronymous from the abstract except for BMLD and DCM, which were both defined.

L. 20-21: BMLD and indicators of the halfway pycnocline are compared to MLD indicators but comparison terms are missing in the sentence. I suggest removing "highly predicted" and insert comparison terms (e.g., "more efficiently predicted").

We agree there was not comparison term. We changed "highly predicted" with "better predicted". L. 24: Remove "as a valuable tool".

We changed the whole sentence, and replaced "as a valuable tool" with "as a potential variable to".

L. 86: Here and elsewhere in the manuscript the term "proxy" is used. Since it has been removed from the title, maybe it can be substituted through all the manuscript.

Thank you for pointing this out. We changed "proxy" with "driver" as suggested by the editor.

L. 160: The algorithm and the maximum angle and cluster analysis methods has been introduce above, thus the present sentence can be modified accordingly (e.g., "The algorithm to identify MLD and BLMD was developed in R (available at [...]) and implements i) the maximum angle method [...] and ii) the cluster analysis [...]").

The sentence has been changed as suggested.

L. 351 and elsewhere: check for occurrence of DMC (instead of DCM). Check also the abbreviations and acronyms used in figures (e.g., "CMd" in Fig. 4 should be DCM).

Thank you very much for pointing this out. We changed all the misspelled acronymous into DCM. L. 366: Number of observations is non-dimensional. To be consistent with the measurement units adopted, the standardization should be done with a length quantity (m). Maybe number of observation multiplied by the observation interval (1 m).

We changed the sentence from "The sum of depth-integrated Chl-a mg m-3 of all profiles was standardized by the number of observations (mg m-3)" to "The sum of depth-integrated Chl-a (mg m-2) of all profiles was standardized by the number of sampling intervals (m)".

L. 414: Maybe "effects" is missing between "physics" and "on primary production". We changed the sentence from "the exclusive investigation of the surface physics on primary production" to "the exclusive investigation of the effects of sea surface processes on primary

**production".**

L. 459: The sentence subject "some of the potential contexts" is placed after the verb "are introduced".

Sentence was corrected as indicated.

**#Reviewer 2**

**Dear Editor.**

I have read the new version of manuscript egusphere-2022-140 by Zampollo A. et. al., now entitled: The mixed layer depth below the pycnocline (BMLD) as an ecological indicator of subsurface chlorophyll-a.

I acknowledge improvements from the previous version I reviewed (was first version, sorry I missed an intermediate version), most following the requests of reviewers. Again, I appreciate the author's effort to provide a systematic characterization and a statistical in-depth analysis from a large dataset and to develop methodological tools. Although I have some concerns on the scope and methodology of the ms, I feel the overall outcomes are valuable and the ms deserves publishing after dealing with some issues I consider minor at this stage.

My first general concern involves the definition of BMLD as 'the depth at which the pycnocline ends and deep mixing develops down to the seabed' (sec.1.3). In my view, this definition seems strongly linked to the top limit of what is known as the benthic boundary layer (BBL) (e.g. Lueck et. al. 2019, doi:10.1016/B978-0-12-409548-9.11622-7) but there is neither connection nor even mention to the BBL across the manuscript. The authors should explain if BMLD is the same as the top of the BBL, if not what are the differences while if yes a new definition/name for an already understood concept may be unnecessary.

We used the definition of "bottom mixed layer depth" (BMLD) to indicate the density level separating the pycnocline from the bottom mixed layer, the last being defined by Pingree and Griffiths (1977) as a layer where the temperature change is 0.01 °C, and Sharples et al. (2001) as a layer where the density change is -0.02 kg m-3 relative to the closest value to the bed. Furthermore, the BML was used by Palmer et al. (2008), Palmer et al. (2013), Wihsgott et al. (2019), Poulton et al. (2022). On the other hand, the bottom boundary layer (BBL) "refers to a layer flow in the immediate vicinity of the solid sea bottom where the effects of viscosity are significant in determining the characteristics of the flow" (from Zhang (2014), DOI 10.1007/978-94-007-6644-0 134-1). The identification of the BBL would, hence, require data on water velocity, density, and seabed topography to measure the water viscosity near the seabed. Zhang (2014) also writes "in a natural continental shelf", such as the North Sea, "any definition of the BBL structure is not straightforward" and that BBL is "just above the sea bottom" where "there is a homogeneous layer of temperature, salinity and density". Similarly, Trowbridge and Lentz (2018) write that BLL "in the ocean is often stably stratified by temperature and salinity [..], even within 1 m of the seafloor [..], and can also be stably stratified by suspended sediment". They defined the BBL as a boundary layer that "is characterized by turbulent eddies that transport mass, heat and momentum across the streamlines and the density surfaces that are associated with ocean

currents and stratification". Hence, the BBL is measured using a set of Reynolds-averaged equations for mass, which requires data on fluxes of momentum, heat, salt and sediments. For these reasons, we recognize differences between BMLD and BBL, although the two are surely interacting in shelf waters. Therefore, we included a section where we describe the potential interactions between BMLD and BBL in the Discussion (lines XX-XX) and we clarified the definition of BMLD in the introduction (1.3 section).

Moreover, Palmer et al. 2013 investigated the physical oceanography of Jones Bank (Celtic Sea) by measuring several physical variables, including the bottom mixed layer and the bottom boundary layer as two different conditions.

Below we summarized the differences between BMLD and BLL and the changes made in the paper:

- BMLD is defined on density profiles (Sharples et al. 2001), BLL is defined using horizontal and vertical speed components (u,v,w) (Trowbridge and Lentz, 2018)
- BMLD is the base of the pycnocline, and can distribute close to the sea surface by following the pycnocline and bottom mixed layer vertical distribution, BLL distributes close to the seabed, in stably stratified layers even within 1 m of the seafloor (Trowbridge and Lentz, 2018).
- BBL and BMLD can interact with each other in shelf waters since BBL is characterized by "turbulent eddies that transport mass, heat and momentum across the streamlines and the density surfaces that are associated with ocean currents and stratification" (Trowbridge and Lentz, 2018).
- As mentioned in the comment below about the conclusions on BMLD=DMC ("I am a bit unsure [..]", we added in Section 1.3 details and references on the use of BMLD to justify the analyses of this paper (more details below).

**I also find a bit misleading the use of "8 density levels DL" to relate to DCM. Some of these only accounts for different methodologies to compute the same thing (MLD, pycnocline depth and BMLD which may be the BBL top).**

To avoid misunderstanding, we stated that DLs are "The depths detailing the density structure in the water column are defined here as density levels (DLs)."

We deleted "eight" before "density levels" across the whole paper. In section 1.3, we mentioned that we investigated four different structures of the density profiles "[...] (MLD, halfway pycnocline depth, BMLD, and maximum frequency buoyancy) are analysed using [...]", and we specified in Section 2.6 that "In this study, we investigate the use of the surface mixed layer depth (MLD0.01, MLD0.02, MLD), the maximum squared buoyancy depth (Max N2), halfway pycnocline and bottom mixed layer depths (HPD0.01-BMLD, HPD0.02-BMLD, HPDMLD-BMLD, and BMLD) to derive the vertical distribution of Chl-a".

I am a bit unsure about the conclusions on DCM and density levels. BMLD and mid-pycnocline shows better linkage with DCM than with MLDs so the authors conclude that MLD metrics are weak predictors of DCM (I.406). I guess the reason is the diverse pycnocline shapes and extent that may occur for any given MLD. Moreover, timescales for physics and phytoplanckton dynamics are different, so I would expect spread on chlorophyll profiles characteristics for very similar density profiles and hence I would not expect that any of the DLs proposed should tightly match as predictor of DCM. Aligned with this, I am not sure on the value of evaluating whether DCM=BMLD (or whatever DL, I.261) besides the distribution of MLD-DL also shown in Fig.5. I think that the authors should elaborate on these issues further.

Thank you for bringing this up. We recognize that this paper does not describe the relationship between DCM and DLs under different hydrodynamic conditions and phytoplankton dynamics, and that the time scales of the processes do not necessarily overlap to each other. However, the Chl-a profiles are likely to change in accordance to the density profile (e.g. Carranza et al., 2018) and the paper aimed not only to investigate DCMs, but also the overall vertical distribution of Chla (section 3.2). The association of MLD with phytoplankton has been described in the literature over and regardless the temporal succession of events defining both the physics and phytoplankton aggregations in the water column (see section 1.2). Therefore, the aim of this paper is instead to investigate at which extent the BMLD can inform on the vertical distribution of DCMs in shelf temperate waters during summer, and showing that BMLD is actually returning information on the vertical distribution of Chl-a maxima, independently from the hydrodynamic conditions, pycnocline stability and phytoplankton phenology status.

However, we agree that the temporal component was not well described in the discussion, although it was taken into consideration during the formulation of the research questions. Therefore, we edited the sections 4, 4.1, 4.2 and the conclusions to give a context on the temporal component while discussing the coincidence of DCM at any DL:

Introduction 1.3: Lines 137-140, "Further scrutiny was applied to BMLD to investigate to which extent the BMLD can inform on the vertical distribution of DCMs in temperate shelf waters during summer, regardless of any phytoplankton dynamic (cell's light history regulating photoacclimation) or physical conditions of the water column (e.g. stability)."

Discussion: Lines 474-487, We agree that the small association of MLD with DCM might be related to the many other factors, defined with the phytoplankton dynamic and succession of physical conditions in the water column. Therefore, we specified that:

"It is worth noting that the comparison between any DL and DCM was made independent of the time scales at which physical processes and phytoplankton dynamics develop, which differ from each other and do not necessarily overlap. Therefore, the association of any DL with DCM (e.g. BMLD=DCM) was investigated under different physical (e.g. water column stability) and biological conditions (e.g. cell's light history regulating photoacclimation) which are likely to be responsible for the unexplained variance reported for each linear comparison in Figure 4. As an example, the small association of DCMs with all the investigated surface mixed layers' indicators (MLD0.01, MLD0.02 and MLD, Table 3) can relate to temporal aspects of the phytoplankton dynamic and physical data set (e.g. multiple data collection within oligotrophic surface waters in stably stratified conditions after spring blooms) at the time of sampling. Hence, the association between any DL and DCM would vary depending on the progression of events defining the profiles of ChI-a and density. Here, we discussed the location of DCMs in regard to MLD, HPD, BMLD and Max N2, considering the potential physical conditions and phytoplankton dynamics at the sampling time (such as water column stability, light history exposure and turbulence) as possible drivers of the resulting associations."

Conclusion: Lines 635-645, "The extent to which subsurface Chl-a maxima distribute in the proximity of any density level was investigated aside from any variable controlling for the progression of

events affecting the physics and biological dynamics of the water column (e.g. vertical Chl-a shape or water column stability) at the sampling time. Hence, the extent of variability retrieved from each comparison (e.g. DCM close to BMLD) is most likely related to the different conditions under which the water columns were investigated, such as the vertical distribution of Chl-a (shapes), nutrients availability, stability of the water column (transition from either stratified to mixed condition or *vice versa*), tidal phase, grazing factors, phytoplankton dynamics (e.g. cell's light history, species composition and competition)."

Moreover, this study can set the basis to develop further questions that would investigate the vertical distribution of Chl-a (shapes), DCM and density features across time when the stratification is set (permanently stratified waters) and ebb/flood cycle increases the distance between MLD and BMLD (enlarge the pycnocline), or when internal waves occur. Hence, we agree that comparing any DL to DCM is as simple as needed in some research field where the physics and phytoplankton dynamics are leaved out, such as deriving the vertical distribution of Chl-a from satellite samplings (Lavigne et al. 2015, 10.5194/bg-12-5021-2015).

Overall, a first description of the vertical distribution of MLD and BMLD can be important to understand whether there are patterns in the vertical distribution of Chl-a under different environmental conditions, e.g. concentrations below MLD in polar regions (Ardyna et al. 2013, 10.5194/bg-10-4383-2013) or below BMLD in coastal waters (< 45 m bathyemtry) (in situ data collected in FoF), close to BMLD in shelf waters (see section 1.1, e.g. Durán-Campos et al., 2019). Since a few methods are described to retrieve BMLD (threshold - Sharples et al. 2001, Wihsgott et al. 2019), the aim of this paper is also to return a useful tool to extract BMLD from high resolution density profiles and potentially state some questions for further investigations (e.g. those mentioned in this reviewer's comment). Therefore, an integrated approach of the threshold method and the maximum angle method (Chu and Fan, 2011) is described and an example on the BMLD's use is given.

Regarding the discussion, climate change and offshore manmade structures are addressed very broadly, highlighting the importance of understanding primary production in a changing environment, but the usefulness of the developed methodology and approach is unclear to me. As mentioned in the last comment made at the end of the document, we clarified how the use of BMLD is useful to investigate potential physical changes related to climate change and man-made structures. We believe that the reviewer referred to the whole method (MLD and BMLD) by writing "developed methodology and approach". However, we want to focus on the reasons behind using BMLD in further studies, as a complementary indicator of the pycnocline position together with MLD and as a driver of subsurface primary production. Moreover, the intention is to shortly summarize the main variations caused by climate change and man-made structure in relation to BMLD/deep mixing processes. We listed below the usefulness of BMLD (and hence the supply of a function/method to retrieve it) in these two contexts:

- Identify the halfway pycnocline depth (HPD), and hence having three indicators of the pycnocline instead of only one, the (surface) mixed layer depth.
- Measures variations in BMLD caused by changes in the deep mixed layer (e.g. changes in the stratification strength due to climate change or increase of the mixing downstream of the turbine foundation)

- Investigate variations in Chl-a abundance, vertical distribution and community composition due to changes in the vertical distribution of BMLD, and its distribution in relation to other factors (e.g. euphotic depth and nutricline).
- Investigating whether grazers, fish or seabirds uses the pycnocline (variations in density throughout the water column) to detect the vertical distribution of food resources, and whether the variation of MLD or BMLD might affect their foraging success.

**Specific Comments:**

**I.18. BMLD should not be explicitly referred in the abstract without providing a definition of the concept, as is done in line 90.**

We added the following sentence "(BMLD: depth between the end of the pycnocline and the below mixed layer)". At line 90 was provided the definition: the mixed layer depth below the pycnocline.

**1.35. not sure about the relevance of seasonality of tide cycles, but I miss a word on the seasonal heating-cooling cycle.**

The seasonal cycle was suggested by a previous reviewer in the second round of revision, and a reference was provided for that (Müller et al. 2014, 10.1007/s10236-013-0679-0). However, the sentence wants to point out that the structure of the stratified water column is highly influenced by the strong daily and biweekly variations in the tidal current (cycles), which ultimately influence the thickness/vertical distribution of the pycnocline's limits and the nutrient-enriched fluxes into the surface waters. For this reason, the seasonal variation of e.g. the lunar component M2 of 10% is not relevant here, and it was deleted from the sentence in this new version. The sentence was also improved to specify that "The vertical distributions of the spring-summer stratification in the water column fluctuate in time and space by the modulation of daily and biweekly strong tidal cycles". Moreover, as indicated by this comment, we included the seasonal heating-cooling cycle in lines 38-41: "The seasonal heating-cooling cycle of the water column regulates the stratification in temperate shelf waters, where the intensified solar radiation in spring-summer increases the difference of temperature and salinity between surface and deep waters and develops a pycnocline dividing surface from deep mixed waters". The seasonal heating-cooling cycle originates the seasonal stratification instead of determining a small temporal scale variation, where BMLD is more relevant. On the other hand, the seasonal heating-cooling cycle is relevant to describe the system where the study is located, and therefore it is added before mentioning the daily/biweekly variation due to tidal cycle.

I.61. It is said that 'the use of MLD is motivated in oceanic sites where the deepest limit of the pycnocline is difficult to draw'. I disagree, the seasonal pycnocline transitions progressively into the permanent thermocline, but it is not difficult to draw a limit, just needed to establish a criteria. This sentence wanted to say that oceanic density profiles can report a smoother rate of change between the pycnocline and the bottom mixed layer (at BMLD). On the other hand, shelf waters are characterized by surface and deep physical processes that make the pycnocline thinner than those in oceanic sites and allow a clearer identification of MLD and BMLD at depths with significant changes in density. Since we wanted to shortly motivate the use of MLD in oceanic sites and introduce the investigation of BMLD in shelf waters, we changed the sentence into "Although the use of MLD is motivated in oceanic sites where the surface processes drive most of the variations in primary production, the biological processes of shelf waters are equally driven by the

physical processes above and below the pycnocline that define the nutrient distribution in a more restricted space. Hence, the identification of the upper and below limits of the pycnocline may improve the understanding of the processes defining the primary production in shelf waters.".

1.79. Interestingly, Kara et.al did not applied a simple threshold but developed an algorithm that involves a previous transformation of the profile (providing a much better result).

We agree with the reviewer, this reference is not entirely correct. We chose Kara et al. 2000 because they listed in table 1 several authors and criteria to measure MLD from the sea surface temperature. We deleted the reference since the threshold method is widely adopted in the scientific community and specific references are only reported to justify the threshold value (as we did in section 2.4 (lines 281-283).

I.480 it is said that 'the role of climate change in increasing stratification is likely to affect the distribution of BMLD and the upward fluxes'. I understand that the main controls of the BBL (hence BMLD) are tidal currents, which will not vary due to climate change. The authors should elaborate further

We addressed this comment together with another comment from the same reviewer "Regarding the discussion, climate change and offshore manmade structures are addressed very broadly [..]". In this section, we mentioned the potential uses of BMLD in future studies. Hence, we listed some of the contexts in which BMLD could be advantageous by referring to:

- Identify the halfway pycnocline depth (HPD), and hence having three indicators of the pycnocline instead of only one, the (surface) mixed layer depth.
- Measures variations in BMLD caused by changes in the deep mixed layer (e.g. changes in the stratification strength due to climate change or increase of the mixing downstream of the turbine foundation)
- Investigate variations in Chl-a abundance, vertical distribution and community composition due to changes in the vertical distribution of BMLD, and its distribution in relation to other factors (e.g. euphotic depth and nutricline).
- Investigating whether grazers, fish or seabirds uses the pycnocline (variations in density throughout the water column) to detect the vertical distribution of food resources, and whether the variation of MLD or BMLD might affect their foraging success.

These points have been better described in section 4.3.

---

## Author Response (AR4)

Dear Prof. Emilio Marañón,

Thank you for providing valuable comments and supporting the publication of this paper with all the reviewers. We have reviewed the manuscript now titled "The bottom mixed layer depth (BMLD) as an indicator of subsurface chlorophyll-a distribution" by addressing the two following comments:

"please consider the following minor revisions when preparing the final version of your paper:
- the title could be more complete. For instance, "The bottom mixed layer depth (BMLD) as an indicator of subsurface chlorophyll-a distribution", or "The bottom mixed layer depth (BMLD) as an indicator of the subsurface chlorophyll-a maximum""

Thank you for suggesting a change in the title. We agreed to change the title into "The bottom mixed layer depth (BMLD) as an indicator of subsurface chlorophyll-a distribution".

"- As noted by the Reviewer, there are some inconsistencies in the labelling of different sections, figures, etc. due to changes in structure through the revision process. This must be carefully revised before final submission."
We corrected the numbers of sections, the number of figures in Appendix A and in the manuscript.

Moreover, while revising the manuscript, we also recognized the importance of adding the following sentence in this final version:

Section 2.3, lines 197-199: "It is important to notice that this method does not determine whether the water column is stratified, and it can be applied to profiles exhibiting a pycnocline described by high-resolution, equally distant observations."

We have also reviewed the introduction (Lines 41-55, Lines 120-125) and a few more changes in the Discussion section "climate change".

We changed "upper" and "lower" in Figure 2 with "surface" and "bottom" to be consistent with the terminology used by Wihsgott et al., 2019, Palmer et al., 2013, Sharples et al. 2001.

Sincerely thank you.

Arianna Zampollo